# Tissue-aware interpretation of genetic variants advances the etiology of rare diseases

Chanan M Argov [1], Ariel Shneyour [1,6], Juman Jubran[1,6], Eric Sabag[1], Avigdor Mansbach[1], Yair Sepunaru [1], Emmi Filtzer[1], Gil Gruber[1], Miri Volozhinsky[1], Yuval Yogev [2], Ohad Birk[2,3], Vered Chalifa-Caspi[4], Lior Rokach [5] & Esti Yeger-Lotem [1,3]✉

## Abstract

**Pathogenic variants underlying Mendelian diseases often disrupt the normal physiology of a few tissues and organs. However, variant effect prediction tools that aim to identify pathogenic variants are typically oblivious to tissue contexts. Here we report a machine-learning framework, denoted "Tissue Risk Assessment of Causality by Expression for variants" (TRACEvar, https:// netbio.bgu.ac.il/TRACEvar/), that offers two advancements. First, TRACEvar predicts pathogenic variants that disrupt the normal physiology of specific tissues. This was achieved by creating 14 tissue-specific models that were trained on over 14,000 variants and combined 84 attributes of genetic variants with 495 attributes derived from tissue omics. TRACEvar outperformed 10 well-established and tissue-oblivious variant effect prediction tools. Second, the resulting models are interpretable, thereby illuminating variants' mode of action. Application of TRACEvar to variants of 52 rare-disease patients highlighted pathogenicity mechanisms and relevant disease processes. Lastly, the interpretation of all tissue models revealed that top-ranking determinants of pathogenicity included attributes of disease-affected tissues, particularly cellular process activities. Collectively, these results show that tissue contexts and interpretable machine-learning models can greatly enhance the etiology of rare diseases.**

**Keywords** Genomic Medicine; Variant Interpretation; Machine Learning; Tissue-selectivity; Variant Effect Prediction
**Subject Category** Genetics, Gene Therapy & Genetic Disease

## Introduction

Mendelian diseases affect hundreds of millions of individuals worldwide, are often incurable, and have limited treatment options (Nguengang Wakap et al, 2020). Rare Mendelian diseases pose even a bigger challenge due to the small number of patients and the complexity of discerning a pathogenic variant out of thousands of candidate genetic variants in a patient (McCarthy and MacArthur, 2017). Consequently, genetic diagnosis is successful in 25–60% of the cases (Chong et al, 2015; The 100,000 Genomes Project Pilot Investigators et al, 2021). The development of treatment options for rare diseases requires further understanding of disease processes, including the molecular mechanisms by which pathogenic variants lead to disease phenotypes. Yet, these mechanisms remain to be elucidated even for well-studied Mendelian diseases (Chong et al, 2015; Goedert et al, 2017; Hernandez et al, 2016; Holmans et al, 2017; Huttlin et al, 2017; Li and Zhang, 2017; Moaven et al, 2015).

Many methods and tools were developed in the last two decades to discern pathogenic variants in protein-coding genes. Main variant effect prediction tools harnessed evolutionary, functional, and genomic attributes of variants, such as sequence conservation (Davydov et al, 2010; Kumar et al, 2009; Margulies et al, 2003; Pollard et al, 2010; Reva et al, 2011; Schwarz et al, 2014), the predicted impact of the aberration on protein function (Adzhubei et al, 2013), and variant frequency in the population (Bosio et al, 2019). Additional tools combined multiple attributes, as done by REVEL (Ioannidis et al, 2016) and the CADD agglomerative database (Rentzsch et al, 2021). Recent methods for variant effect prediction, such as EVE (Frazer et al, 2021) and ESM1b (Rives et al, 2021), applied deep learning models to protein sequence databases, or combined them with protein structures predicted by AlphaFold2 (Jagota et al, 2023).

A different class of tools was designed to predict disease genes rather than pathogenic variants. This has been achieved by using gene attributes unavailable for variants, at the cost of disregarding the actual genetic aberration. Gene prioritization tools harnessed omics profiles, mostly transcriptomics, as well as molecular interactions, functional information, gene annotations, and constraint genetic variation (Aerts et al, 2006; Karczewski et al, 2020; Kumar et al, 2018; Tranchevent et al, 2016; Yao et al, 2018). Additional tools also exploited phenotypic classification of patients by the Human Phenotype Ontology (HPO), and prioritized candidate variants or genes by comparing their attributes to the attributes of known phenotype-associated genes (Cipriani et al, 2020; Deelen et al, 2019; Kumar et al, 2018).

[1]Department of Clinical Biochemistry and Pharmacology, Faculty of Health Sciences, Ben-Gurion University of the Negev, Beer Sheva 84105, Israel. [2]Morris Kahn Laboratory of Human Genetics and the Genetics Institute at Soroka Medical Center, Faculty of Health Sciences, Ben Gurion University of the Negev, Beer Sheva 84105, Israel. [3]The National Institute for Biotechnology in the Negev, Ben-Gurion University of the Negev, Beer Sheva 84105, Israel. [4]Ilse Katz Institute for Nanoscale Science & Technology, Ben-Gurion University of the Negev, Beer-Sheva 84105, Israel. [5]Department of Software & Information Systems Engineering, Faculty of Engineering Sciences, Ben-Gurion University of the Negev, Beer Sheva 84105, Israel. [6]These authors contributed equally: Ariel Shneyour, Juman Jubran. ✉E-mail: estiyl@bgu.ac.il

Variant effect prediction and gene prioritization tools have employed computational techniques ranging from simple calculations to sophisticated machine-learning (ML) methods (Bosio et al, 2019; Kumar et al, 2018; Li et al, 2020; Quang et al, 2015), including supervised ensemble methods, such as ClinPred (Alirezaie et al, 2018), BayesDel (BayesDel_addAF and BayesDel_noAF)(Feng, 2017), and REVEL (Ioannidis et al, 2016), and unsupervised deep learning models, such as EVE and ESM-1b. Yet, most tools offered limited insight into their decisions or into the downstream functional consequences of the sequence variation. Thus, disease processes have often remained hidden.

An important aspect of Mendelian diseases is their tendency to manifest clinically in selected tissues, as demonstrated by inherited kidney diseases and familial neurological disorders (Huang et al, 2014), and reviewed in (Hekselman and Yeger-Lotem, 2020). Some prioritization methods utilized tissue omics profiles to ascertain the relevance of candidate genes and variants in disease-affected tissues (Cummings et al, 2020; Cummings et al, 2017; Yao et al, 2018), or produced catalogs of tissue-relevant genes (Simonovsky et al, 2023; Somepalli et al, 2021). We recently reported a tissue-aware ML framework denoted "Tissue Risk Assessment of Causality by Expression" (TRACE) that predicts disease genes in a tissue context by the likelihood of the respective disease to manifest in that tissue (Simonovsky et al, 2023). The TRACE framework was unique in utilizing gene attributes (denoted in ML as features) matching known tendencies of disease genes, such as their tendency for preferential expression in normal tissues that manifest pathology (Lage et al, 2008; Sonawane et al, 2017), for tissue-enhanced molecular interactions (Barshir et al, 2014; Basha et al, 2020; Chen et al, 2021; Marbach et al, 2016), for tissue-selective compensatory mechanisms (Barshir et al, 2018; Jubran et al, 2020), and for involvement in preferentially active cellular processes (Sharon et al, 2022). TRACE outperformed other gene prioritization methods in cross-validation and when tested on data from rare-disease patients. Yet, TRACE was applicable to eight tissue contexts, and, like other gene prioritization tools, was oblivious to the actual genetic aberration.

Here we present the "TRACE to variants" (TRACEvar) framework that predicts pathogenic variants in tissue contexts. TRACEvar dramatically extended TRACE by making it applicable to genetic variants rather than genes. This was achieved by combining 84 variant-specific features from CADD with 495 tissue-specific features from TRACE, and by providing variants with tissue contexts. In cross-validation assessments of over 14,000 genetic variants in 14 tissue contexts, TRACEvar showed high accuracy (median auROC = 0.92; median auPRC = 0.23, expected 0.018). When applied to a test set of recently detected variants, TRACEvar outperformed 10 well-established variant effect prediction methods. Lastly, when tested on candidate variants of 52 rare-disease patients, in more than half of the cases TRACEvar ranked the verified pathogenic variant among the top 10% candidate variants of that patient.

Another important advantage of TRACEvar is the interpretability and explainability of its models. These allowed us to illuminate likely pathogenicity mechanisms and disease-related processes for specific pathogenic variants, and to highlight determinants of pathogenicity that were common across variants and tissues. To make TRACEvar accessible to the scientific community, we developed a publicly available TRACEvar webtool (https://netbio.bgu.ac.il/tracevar/) that does not require any programming skills. The webtool offers prioritization and interpretation of user-uploaded variants in 14 tissue contexts.

## Results

### Overview of TRACEvar ML framework

TRACEvar aimed to assess the pathogenicity of variants in specific tissues. For this, we created a features dataset that contained 84 variant-specific features and 495 tissue-specific gene features (Fig. 1, "Methods"). The variant-specific features were extracted from CADD and pertained, for example, to the conservation of the mutated region and the predicted impact of the mutation on protein function (Rentzsch et al, 2021). The tissue-specific features were extracted from TRACE (Simonovsky et al, 2023), and integrated heterogeneous omics data. These included transcriptomes of 54 normal adult tissues (The GTEx Consortium, 2020) and seven fetal tissues (Cardoso-Moreira et al, 2019), tissue eQTLs (The GTEx Consortium, 2020), experimentally detected protein–protein interactions (PPIs) (Basha et al, 2015; Basha et al, 2018; Ziv et al, 2022), paralogous genes (Jubran et al, 2020) and gene ontology (GO) biological process annotations (Huntley et al, 2015). We particularly employed tissue-specific features that reflect known tendencies of disease genes in normal tissues. Some of those features were derived from a single type of data, reflecting for example, the preferential expression of the variant-containing gene in different tissues (Barshir et al, 2014; Lage et al, 2008; Sonawane et al, 2017). Other features integrated multiple types of data. For example, tissue transcriptomes and GO biological processes were combined to derive the differential activity of biological processes involving the variant-containing gene in different tissues (Sharon et al, 2022). For simplicity of presentation, we grouped the 84 variant-specific features into 11 groups, such as the "pathogenic score" group that included pathogenicity scores by CADD, PolyPhen-2 and SIFT, and the "variant location" group that included features pertaining to the location of variation within the gene or protein. Likewise, we grouped the 495 tissue-specific features into six main groups (Appendix Table S1; Fig. 1B). For example, the "expression time" group contained features pertaining to gene expression at different time points during development, and the "PPI group" contained PPI-related features. Altogether, our dataset contained 579 features, most of which were tissue-specific (Appendix Table S1).

To support supervised ML, we created a dataset of variants that were labeled as pathogenic or benign in tissue contexts (Fig. 1A). For this, we extracted data of pathogenic and benign variants from ClinVar (Landrum et al, 2016). We used their stars method to enhance the reliability of the dataset ("Methods"). Pathogenic variants were limited to causal variants for hereditary diseases that were recorded in the Online Mendelian Inheritance in Man (OMIM) catalog (Amberger et al, 2009) and were known to manifest clinically in certain tissues (Hekselman et al, 2022) ("Methods"). We labeled these variants as pathogenic in their disease-affected tissue and benign in other tissues. For example, variants of the gene TNNT2 that were causal for cardiomyopathy were labeled as pathogenic in heart tissue and benign in other tissues (Kamisago et al, 2000). Altogether, we labeled 18,634 variants (3492 genes) in 14 tissues. Of these, 2908 variants (616 genes) were labeled as pathogenic in at least

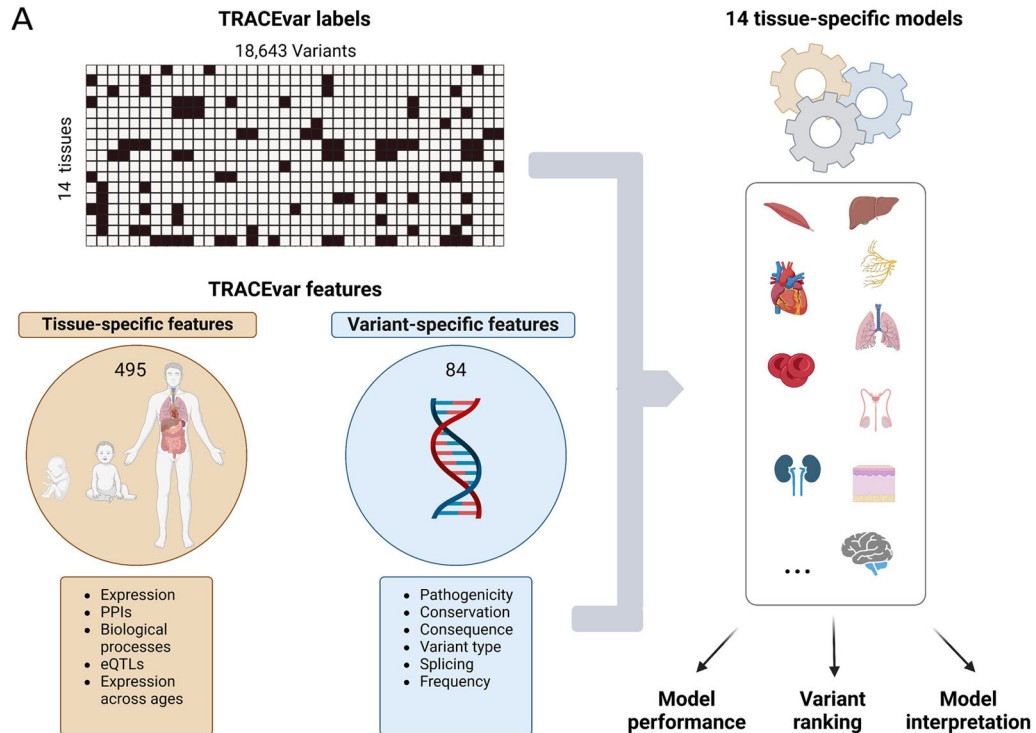

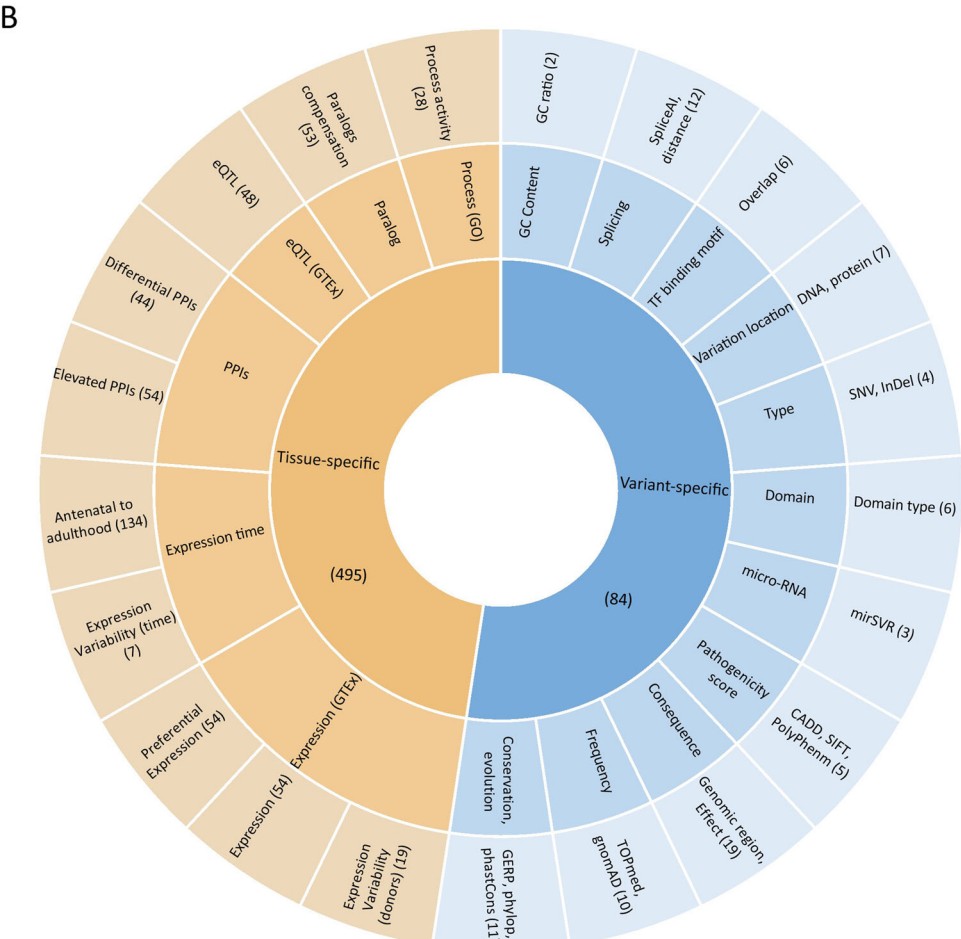

Figure 1. Overview of TRACEvar machine-learning approach.

(A) TRACEvar assesses the pathogenicity of genetic variants in tissue contexts, based on tissue-specific labeling of variants, tissue omics, and variant-specific features. TRACEvar labels included a matrix of 18,634 genetic variants (columns) and 14 tissues (rows); black entries represent genetic variants that are pathogenic for a disease that manifests clinically in the respective tissue. TRACEvar features consisted of 495 tissue-specific features that pertain to adult and developing tissues, and 84 variant-specific features. The labels and features were used to train and test 14 tissue models that classify variants as pathogenic or benign in a tissue-specific manner. The resulting models were used to rank candidate pathogenic variants, and to illuminate the features that led to the model's decision. (B) The types of features used in TRACEvar. Tissue-specific features (orange) included molecular attributes of genes and were grouped by main type of data (middle circle) and by type (outer circle). Variant-specific features (blue) included attributes of variants and were grouped by type (middle circle) and source of data (outer circle). Total number of features per type appears in parenthesis. PPI protein–protein interaction, eQTL expression quantitative trait loci.

one tissue, with a median frequency of 1.7% pathogenic variants out of the total number of variants per tissue (Appendix Table S2 and Appendix Fig. S1). Each variant was associated with its variant-specific features and with the tissue-specific features of its corresponding gene.

Next, we tested whether TRACEvar could classify pathogenic versus benign variants in the context of specific tissues. For classification we used gradient boosted machine (GBM), a powerful decision tree-based ML method (Friedman, 2001). As proof of concept, we applied TRACEvar to predict the pathogenicity of variants in the genes TNNT2 and AHDC1 in the context of six tissues (Fig. 2A,B). The gene TNNT2 encodes troponin T2, a cardiac isoform of the tropomyosin-binding subunit of the troponin complex, which regulates muscle contraction in response to alterations in intracellular calcium ion concentration and is expressed specifically in the heart. Our labeled dataset contained 18 TNNT2 variants, of which 10 were benign and eight were pathogenic and causal for cardiomyopathy (OMIM #601494, #612422, and #115195). Therefore, we applied TRACEvar to predict the pathogenicity of TNNT2 variants in heart tissue and in five unaffected tissues. Per tissue, we trained a separate GBM model on all variants in our labeled dataset except for TNNT2 variants, and then applied the resulting model to predict the pathogenicity in that tissue of the 18 TNNT2 variants ("Methods"). The 10 benign TNNT2 variants obtained a low TRACEvar pathogenicity score in all tissue models (Fig. 2A). The eight pathogenic TNNT2 variants obtained low TRACEvar pathogenicity scores in models of unaffected tissues, and high TRACEvar pathogenicity scores in the heart model (Fig. 2A).

Next, we applied TRACEvar to predict the pathogenicity of variants in the gene AHDC1. AHDC1 encodes a transcription factor that is a key regulator of early epithelial morphogenesis (Collier et al, 2022), and is widely expressed across adult tissues GTEx portal (GTEx Consortium, 2024). Our labeled dataset contained 23 benign variants of AHDC1 variants, and three variants that are causal for mental retardation (Xia-Gibbs syndrome, OMIM #615829) and hence labeled as pathogenic in brain tissue. We applied TRACEvar to predict the pathogenicity of the 26 AHDC1 variants in brain tissue and five other tissues, as described above. The 23 benign variants of AHDC1 obtained low TRACEvar pathogenicity scores in all tissue models (Fig. 2B). The three pathogenic variants scored high in the whole-brain tissue model and low in all other tissue models (Fig. 2B). These examples demonstrate that TRACEvar can successfully distinguish between benign and pathogenic variants in tissue contexts.

## Large-scale assessment of TRACEvar in 14 tissue contexts

To test TRACEvar at a large scale, we trained and tested ML models for 14 tissue contexts, including whole brain and four brain subregions (cortex, cerebellum, basal ganglia, and spinal cord),

heart, kidney, liver, lung, skeletal muscle, skin, testis, tibial nerve, and whole blood. To reduce the risk for data leakage, we divided the dataset of labeled variants into training and test sets ("Methods"). The training set contained 14,543 variants (2500 genes) that were annotated by the year 2022. Of those, 2360 variants (527 genes) were pathogenic in at least one tissue, with a median of 313 pathogenic variants (57 genes) per tissue (Appendix Table S2A). The test set contained 2429 variants (1314 genes) that were annotated after the year 2022, while limiting its gene overlap with the training set ("Methods"). Of those, 490 variants (256 genes) were labeled as pathogenic in at least one tissue, with few tens of pathogenic variants from a similar number of genes per tissue (Appendix Table S2B and Appendix Fig. S1).

We applied six ML classification methods, including K-nearest neighbors (KNN), logistic regression (LR), and four decision forest methods: GBM, random forest (RF), extreme gradient boosting (XGB), and adaptive boosting (AdaBoost). Using the training set, we assessed the performance of each method per tissue via cross-validation, while maintaining similar proportions of pathogenic variants per fold and no gene overlap between the training and validation sets ("Methods"). Next, we estimated the performance of the different models by the average area under the receiver-operating characteristic curve (auROC, Appendix Fig. S2A), and the average area under the precision–recall curve (auPRC, Appendix Fig. S2B), as well as the training and prediction times (Appendix Fig. S2C,D). Decision forest methods had higher fitting time than other methods and achieved better auROC and auPRC (Appendix Fig. S2). Among them, GBM achieved slightly higher auROC (median 0.92) and auPRC (median 0.23, expected 0.018, Appendix Table S2A), and was henceforth selected for TRACEvar implementation. The highest performance was observed for the whole-brain tissue model (auROC = 0.97 and auPRC = 0.75, expected 0.07), which also had the largest number of pathogenic variants. Likewise, the lowest performance was observed for tissue models with low numbers of pathogenic variants, such as certain brain subregions, liver, and lung. Indeed, the number of pathogenic variants and the auPRC were correlated across tissue models (Spearman $r = 0.63$ and $P = 0.018$, Appendix Fig. S2E).

Next, we applied TRACEvar to our test set. The performance of TRACEvar was high across all tissue models (median auROC = 0.98 and auPRC = 0.89, respectively, Appendix Table S2B). Since performance was higher than in the training set, we repeated the performance analysis upon excluding from the test dataset variants in genes with any additional variant in the training dataset. The revised test set contained only few pathogenic variants per tissue model and thus was less representative (Appendix Table S2). Whereas performance was reduced, the number of pathogenic variants in the training dataset and the auPRC remained strongly correlated across tissue models (Spearman $r = 0.9$ and $P = <2e-16$, Appendix Fig. S2F).

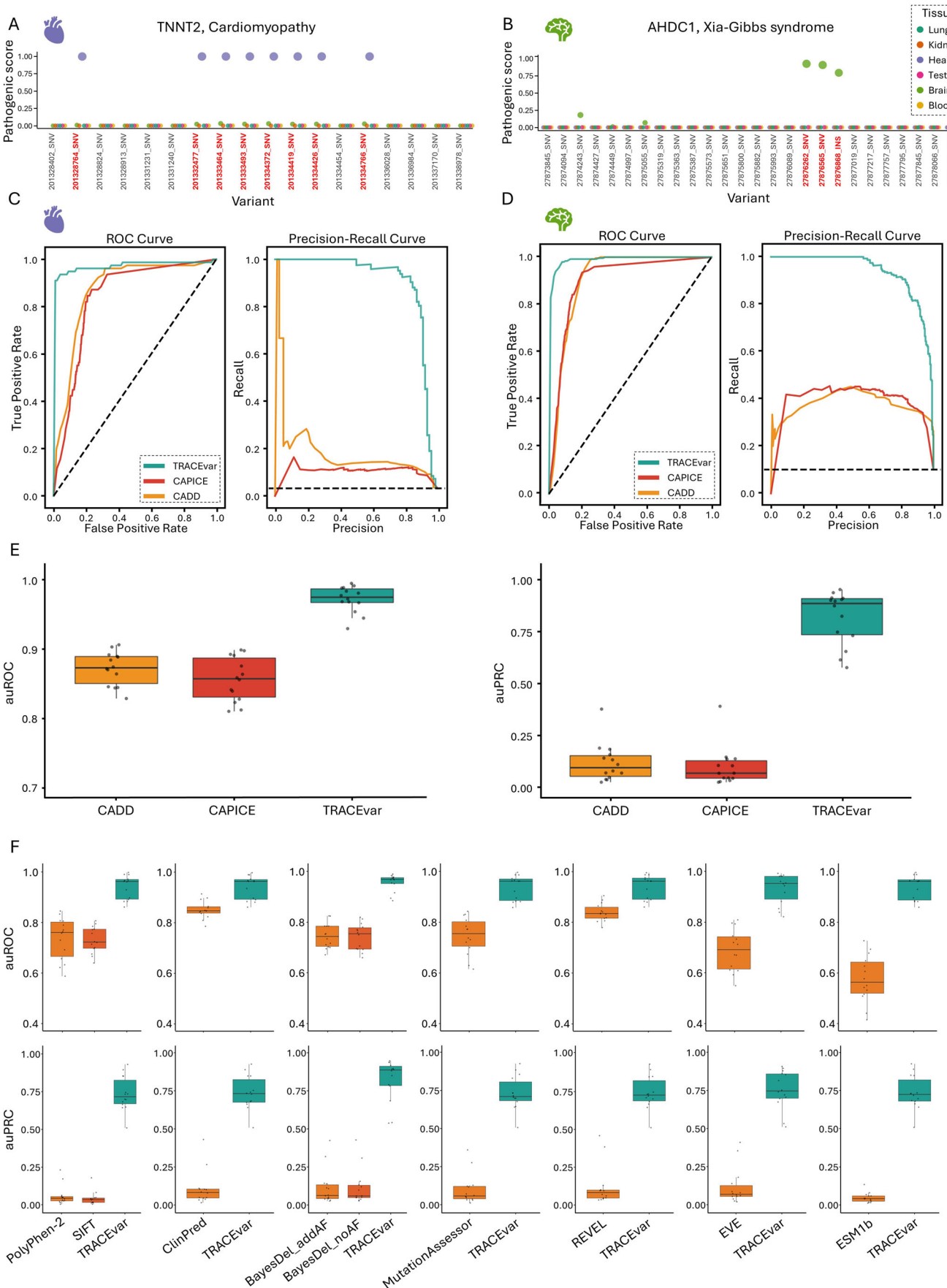

**Figure 2. Assessment of TRACEvar tissue models.**

(A) TRACEvar pathogenicity scores of 18 genetic variants in the gene TNNT2. The variants (x axis) included eight variants that were pathogenic for cardiomyopathy (red labels). Each dot represents the predicted pathogenic probability (y axis) of a variant in six tissues. Benign variants scored low in all tissues. Pathogenic variants scored high only in heart tissue (purple dots) where the disease manifests. (B) Same as (A), for 26 variants in the gene AHDC1, including three variants that were pathogenic for Xia-Gibbs mental retardation syndrome (red labels). Benign variants scored low in all tissues. Pathogenic variants scored highest in brain tissue (green dots) where the disease manifests. (C) Performance of TRACEvar, CAPICE and CADD, when applied to variants from the test set, for diseases that manifest in heart tissue. Performance was measured by auROC (left) and auPRC (right). Black dashed lines represent random guess. (D) Same as (C) for diseases that manifest in brain tissues. (E) Performance of TRACEvar, CAPICE, and CADD in predicting pathogenic variants from the test set per tissue model (n = 14), measured via auROC (left) and auPRC (right). TRACEvar outperformed other methods (TRACEvar versus CAPICE and CADD, adjusted P < 6.7e-5; one-tailed paired Wilcoxon test; The exact P values appear in the Source Data). Boxplot central band indicates median; box limits indicate 25th to 75th percentiles; whiskers indicate 1.5 × interquartile range. (F) Performance of TRACEvar and additional variant effect prediction tools in predicting pathogenic variants from the test set per tissue model (n = 14), measured via auROC (upper row) and auPRC (bottom row). TRACEvar outperformed all other methods. The number of compared variants per panel and the adjusted P value based on one-tailed paired Wilcoxon test were: PolyPhen-2 with 720 variants and P < 6.7e-5; SIFT with 720 variants and P < 6.7e-5; ClinPred with 744 variants and P < 1.2e-4; BayesDel with 1040 variants and P < 6.7e-5; MutationAssessor with 654 variants and P < 6.7e-5; REVEL with 727 variants and P < 6.7e-5, EVE with 321 variants and P < 6.7e-5, ESM1b with 710 variants and P < 6.71e-5. The exact P values appear in the Source Data. Boxplot representation is as described above.

We compared the performance of TRACEvar to a variety of well-established and state-of-the-art variant effect prediction methods: The classical methods SIFT (Kumar et al, 2009), PolyPhen-2 (Adzhubei et al, 2013), MutationAssessor (Reva et al, 2011), and CADD (Rentzsch et al, 2021); the modern ensemble tools ClinPred (Alirezaie et al, 2018), BayesDel (Feng, 2017), REVEL (Ioannidis et al, 2016), and CAPICE (Li et al, 2020); and the unsupervised deep learning models EVE (Frazer et al, 2021) and ESM1b (Rives et al, 2021). In each comparison, we applied the specific method and TRACEvar to variants in the test set that were scored by both (Methods). For CADD and CAPICE the comparison included the entire test set, and the results for the heart and whole brain are demonstrated in Fig. 2C,D. Since the test set included recently annotated variants, it was not likely included in the training of other tools, thereby allowing an unbiased comparison. Across all tissue models, the auROC and auPRC values of TRACEvar were significantly higher than all other methods (adjusted P ≤ 1.2e-4 and P = 6.1e-5, respectively, one-tailed paired Wilcoxon test; Fig. 2E,F). Hence, tissue-specific scoring improves the prediction of variant pathogenicity.

Since the compared methods were not tissue-aware, we turned to assess the performance of a hybrid approach that combined a tissue-unaware variant effect prediction method with a tissue-aware gene prioritization method. To implement the hybrid approach, we used the tissue-unaware REVEL tool and the tissue-aware TRACE method (the latter available for ten tissues, "Methods"). TRACEvar outperformed the hybrid approach in all ten tissue contexts and was significantly better in 8/10 (adjusted P ≤ 0.01, one-tailed paired Wilcoxon test; Appendix Fig. S3). Hence, simultaneously considering variants in tissue contexts is superior to a hybrid approach.

We also tested what would be the impact of using the wrong tissue model. For that, per tissue, we predicted the labels of its variants using each of the models trained on other tissues, and then compared the auROCs between models. Apart from lung, which had equally high auROC in the lung and skin models, the auROC of the correct tissue models were always higher (Appendix Fig. S4). Hence, tissue contexts are not only important in general but also distinct from each other.

Lastly, we tested whether TRACEvar could be used to predict the tissue that is likely affected by a pathogenic variant. For that, we compared the ranks of pathogenic variants in our test set between models of their affected tissue and other tissues. In all 14 tissues,

the median rank of pathogenic variant was highest in the model of their affected tissue (Fig. EV1). In 12/14 tissues, the difference between models of the affected tissue and other tissues was statistically significant (adjusted P ≤ 0.03, one-tailed paired Wilcoxon test). Exceptions included lung, whose model was significantly higher than all other models except skin, and a brain sub-region, whose model was significantly higher than all other models except nerve and another brain sub-region. This analysis suggests that, given a pathogenic variant, TRACEvar could be used to narrow the set of likely affected tissues.

## TRACEvar application to clinical data illuminates disease mechanisms

We next tested the ability of TRACEvar to classify and interpret candidate variants in patients with rare hereditary diseases. For that, we used exome-sequencing data from 52 genetically diagnosed patients, namely patients whose pathogenic variant was identified. Per case, we gathered the list of candidate variants of that patient. All candidate variants were first filtered to remove unlikely candidates, for example alleles that are common in the population ("Methods"). The number of remaining candidate variants per patient ranged between 102 and 1700 (median = 241 candidate variants, Appendix Table S3). 47 patients had one primary disease-affected tissue and five patients had two primary disease-affected tissues and thus were modeled in both, resulting in 57 models pertaining to six tissues (whole-brain, 39 cases; heart, 3 cases; kidney, 1 case; skeletal muscle, 9 cases; skin, 4 cases; testis, 1 case). Next, we created tissue models by training them on the entire dataset of variants ("Methods"). We then applied the model fitting with the patient's affected tissue to the patient's candidate variants, prioritized them according to the model's pathogenicity scores, and extracted the rank of the verified pathogenic variant among the patient's candidate variants ("Methods").

In 54/57 (95%) of the cases, the verified pathogenic variant ranked above the median, and in 46/57 (81%) it was ranked among the top quartile (Figs. 3A and EV2A,B). Notably, in 32/57 (56%) cases, the verified pathogenic variant ranked among the top 10% variants (Fig. EV2A). Next, we compared the rank of the verified pathogenic variant by TRACEvar to the ranks obtained by each of the ten other methods, upon considering only variants that were ranked by both (Dataset EV1). Since patients' cases were solved previously with the help of such tools, we expected that, owing to circularity, other tools would

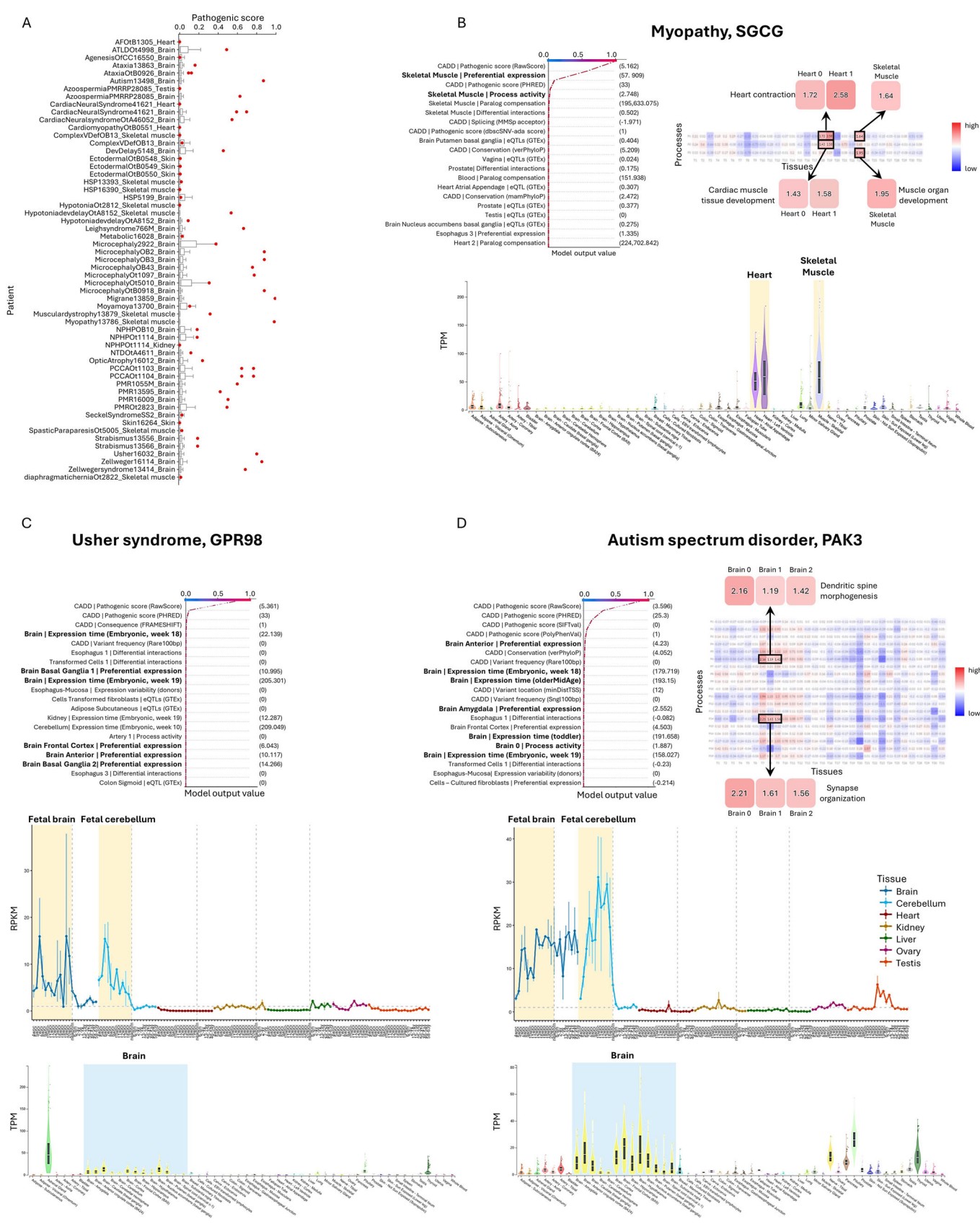

**Figure 3. TRACEvar analysis of clinical cases.**

(A) Box plot summaries of TRACEvar pathogenic scores of variants (x axis) in 57 cases (y axis). Each box represents the scores of all candidate variants of one case, as calculated by the model of the patient's affected tissue. Red dots mark the score of the verified pathogenic variant (one or two per case). Boxplot central band indicates median; box limits indicate 25th to 75th percentiles; whiskers indicate 1.5 × interquartile range. (B) SHAP explanation of the model for a SNV in SGCG that is pathogenic for myopathy. The 20 topmost contributing features are shown, ordered from bottom to top by their increased absolute contribution to the model (i.e., the topmost contributing feature appears first). Of the top six features, four were specific to skeletal muscle tissue. They pointed to preferential expression of SGCG in adult skeletal muscle tissues (bottom panel), and to high preferential activity of processes involving SGCG in skeletal muscle tissue, including "muscle organ development" (heatmap). (C) SHAP explanation of the model for a frameshift deletion in GPR98 that is pathogenic for Usher syndrome. The 20 topmost contributing features are shown. The fourth top feature pointed to high expression of GPR98 in embryonic brain tissue (middle panel), and additional features pointed to preferential expression of GPR98 in adult brain tissues (lower panel). (D) SHAP explanation of the model for a missense SNP in PAK3 that is pathogenic for autism spectrum disorder (ASD). The 20 topmost contributing features are shown. Among them, eight features were related to brain tissues. These features pointed to preferential expression of PAK3 in adult brain tissue (bottom panel), high expression of PAK3 in embryonic brain tissue (middle panel), and high preferential activity of processes involving PAK3 in brain tissues, the topmost of which were "synapse organization" and "dendritic spine development" that are known to play a role in ASD (heatmap; image extracted from (Sharon et al, 2023)). Images of expression in adult tissues, embryonic tissues, and preferential activity of processes, were extracted from the GTEx portal (GTEx Consortium, 2024), from (Cardoso-Moreira et al, 2024), and from ProAct web-server (Sharon et al, 2024), respectively. The SHAP explanation of each case is also available in Dataset EV3.

perform well in detecting the verified pathogenic variant. Nevertheless, TRACEvar performed better than 6/10 tools, was significantly better in 4/10 ($P \le 0.005$, one-tailed paired Wilcoxon test), and performed similarly to the remaining tools (Fig. EV2C).

Unlike other methods, TRACEvar models were interpretable, meaning that the contribution of each feature to the model could be estimated. For that, we used SHAP (SHapley Additive exPlanations) algorithm, which is a game-theoretic method for explaining the prediction of ML models (Lundberg et al, 2020). We next checked whether top-contributing features could illuminate disease-related mechanisms. Below we discuss three patient cases.

Patient 13786 was diagnosed with myopathy, a skeletal muscle disease. The patient had 127 candidate variants in 117 genes. The verified pathogenic variant was a single-nucleotide variation (SNV) in the gene SGCG that encodes gamma-sarcoglycan, a component of a subcomplex of the dystrophin–glycoprotein complex which forms a link between the cytoskeleton and the extracellular matrix of muscle cells. According to gnomAD, SGCG is tolerant to variation (e.g., loss-of-function (LoF) variation observed/expected=0.84, pLI=0), yet variants in that gene are known to cause limb-girdle muscular dystrophy (OMIM #253700). The verified pathogenic variant was successfully ranked first by the TRACEvar skeletal muscle model, and lower by other methods, except BayesDel (Dataset EV1). SHAP analysis revealed that the second most contributing feature to the TRACEvar model was the preferential expression in skeletal muscle tissue, which was indeed high for SGCG as observed using GTEx (Fig. 3B). Another contributing feature was the high activity of SGCG-related processes in skeletal muscle tissue, particularly due to SGCG involvement in the processes "muscle organ development", "cardiac muscle tissue development", and "heart contraction", which were preferentially active in skeletal muscle and heart tissues (Fig. 3B).

Patient 16032 was diagnosed with Usher syndrome, a genetic disorder affecting hearing and vision. The patient had 179 candidate variants in 146 genes. The verified pathogenic variant was a deletion of one nucleotide in the gene GPR98 (ADGRV1) that encodes a G-protein coupled receptor. The gnomAD LoF score of GPR98 was quite low (observed/expected = 0.44). The variant was ranked first by the TRACEvar whole-brain model, and lower by CADD and CAPICE (ranked 5 and 8, respectively, Dataset EV1; scores by other methods were not available). SHAP analysis revealed that the top-contributing features were pathogenicity score and frameshift (Fig. 3C). Notably, 7/20 top-

contributing features were related to GPR98 expression in brain and in early stages of brain development. Indeed, GPR98 was expressed relatively high in adult brain tissues and particulary high in embryonic and newborn brain tissues (Fig. 3C).

The last example is patient 13498, diagnosed with autism spectrum disorder (ASD), that had 242 candidate variants in 201 genes. The verified pathogenic variant was a missense SNP in the gene PAK3 that encodes a protein kinase and plays a role in dendrite spine morphogenesis and synapse formation and plasticity. PAK3 is intolerant to missense variation (gnomAD, observed/expected = 0.32). This variant was ranked second by the TRACEvar whole-brain model, and was ranked lower by other methods, except for CAPICE that ranked it first and BayesDel that ranked it second, Dataset EV1). SHAP analysis of the TRACEvar model revealed the contribution of brain-specific features, with nine brain-specific features among the top 20 features (Fig. 3D). They pointed to the high expression of PAK3 in brain and in embryonic stages of brain development (Fig. 3D), and the high activity of PAK3-related processes in brain tissue, including "synapse organization" and "dendritic spine morphogenesis" (Fig. 3D), consistent with known ASD factors.

To facilitate similar analyses of genetic variants and particularly of data from patients, we developed the TRACEvar webserver (https://netbio.bgu.ac.il/tracevar/), that hosts the tissue models described above. Users can upload to the webserver a list of variants or a VCF file and select a tissue model. The output consists of a prioritized list of the query variants by their TRACEvar pathogenicity scores in the selected tissue. Per variant, the output also contains the SHAP analysis, namely a ranked list of the features that contributed to the model's decision, thereby illuminating its potential pathogenic characteristics.

## TRACEvar reveals determinants of pathogenicity

Lastly, we wanted to answer the following questions: What are the main features that contribute to the pathogenicity of genetic variants in tissue contexts? What is the contribution of tissue-specific versus variant-specific features? To answer these questions, we applied SHAP to each tissue model (Dataset EV2). We demonstrate the results of the SHAP analysis by focusing on the 20 topmost contributing features to the heart and whole-brain tissue models (Fig. 4A). In both models, the two topmost contributing features belonged to the variant-specific group of

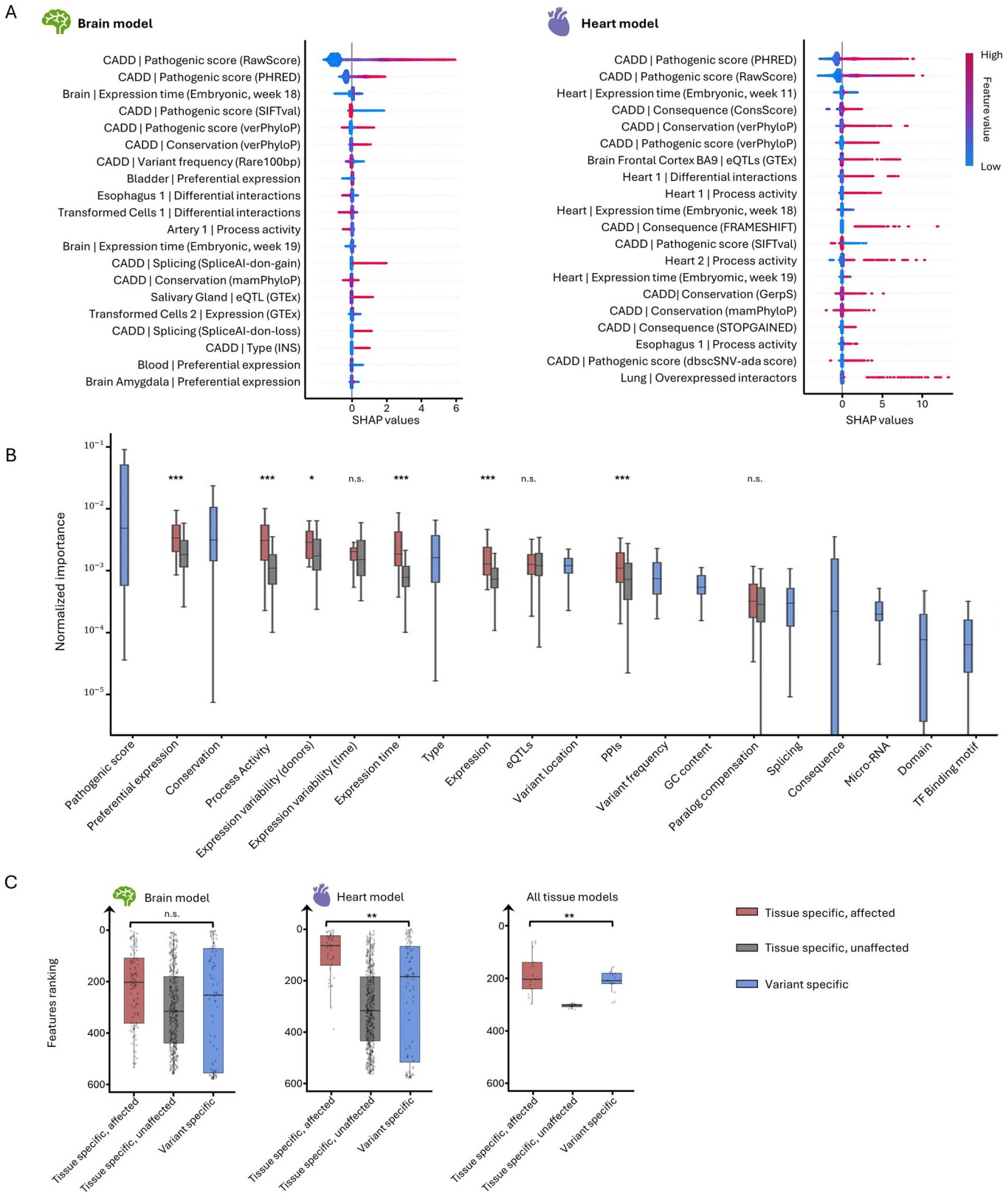

**Figure 4.   Interpretation of TRACEvar models reveals characteristics of pathogenic variants.**

(A) The 20 topmost contributing features of the heart (left) and whole-brain (right) tissue models. Features were ordered from bottom to top by their increased absolute contribution to the model. Per feature, each dot represents the feature value of a different variant (red and blue denote high and low values of the feature, respectively). Dots are spread from left to right by their contribution to disease manifestation in the modeled tissue. (B) The normalized importance ($x$ axis) of 20 different feature groups ($y$ axis) across tissue models ($n = 14$). The nine groups of tissue-specific features were separated into features of disease-affected tissues (red) and unaffected tissues (gray). In 6/9 tissue-specific feature groups, the contribution of affected tissue features was significantly higher than features of unaffected tissues (MW test, adjusted $P$ values: $*P < 0.05$, $***P < e-10$; n.s. = not significant; exact $P$ values appear in the Source Data). Boxplot central band indicates median; box limits indicate 25th to 75th percentiles; whiskers indicate 1.5 × interquartile range. (C) The ranking of variant-specific features (blue), tissue-specific features of the modeled tissue (red), and tissue-specific features of other tissues (gray) for the whole-brain tissue model, the heart tissue model, and across all tissue models. Tissue-specific features of the modeled tissue ranked better than variant-specific features in the heart tissue model (one-tailed MW test, adjusted $P = 0.0009$) and across all tissue models ($n = 14$; one-tailed paired Wilcoxon test, $P = 0.019$), and significantly better than tissue-specific features of other tissues in all models (one-tailed MW test, adjusted $P \leq 8.7e-6$ for whole brain and heart models; one-tailed paired Wilcoxon test, $P = 0.001$ for all tissue model). Boxplot representation is as described above. Feature numbers for the whole brain and heart tissue models: variant-specific features $n = 84$; tissue-specific features of the modeled tissue $n = 102$ and 36, respectively; tissue-specific features of other tissues $n = 393$ and 459, respectively.

pathogenic scores. The third topmost contributing features in both models belonged to the tissue-specific group of expression time, and particulrly to expression time in the modeled tissue (i.e., expression time in heart tissue in the heart tissue model, and expression time in brain tissue in the whole-brain tissue model). This suggests that tissue-specific features of the modeled tissue are important for the models' success.

Next, we turned to the SHAP results of other tissue models, while focusing on the contribution of each feature group to the models. To test if tissue-specific features of the modeled tissue were more important than features of other tissues, we further divided the groups of tissue-specific features accordingly ("Methods"). Then, per model, we scored the importance of each group of features by the average normalized contribution of its features to the model, as estimated by SHAP ("Methods", Fig. 4B). Across models, the topmost contributing group was pathogenic score. Since this group included well-established scoring schemes, its top rank supported the validity of our analyses. The second topmost group was tissue-specific preferential expression, with preferential expression in the modeled tissue being significantly more important than preferential expression in other tissues. This alternating pattern between groups of variant-specific and tissue-specific features continued. The conservation group ranked third, in accordance with the expected deleterious impact of mutations in conserved regions. The tissue-specific groups biological processes, expression variability across donors or time, and expression time, ranked fourth to seventh, and also showed advantage for features of the modeled tissue. This is in accordance with previous studies showing that disease genes tend to be preferentially expressed and involved in process that are preferentially active in disease-susceptible versus unaffected tissues (Barshir et al, 2018; Lage et al, 2008; Sharon et al, 2022; Simonovsky et al, 2019; Simonovsky et al, 2023; Sonawane et al, 2017). A closer inspection of the tissue-specific features that contributed to each model by their types and tissue-of-origin appears in Appendix Fig. S5.

To further assess the contribution of tissue-specific versus variant-specific features per model we compared their ranks. In the whole brain and heart models, for example, tissue-specific features of the modeled tissue ranked better than either variant-specific features and features of other tissues, though only in the heart model the difference was statistically significant (adjusted $P = 1e-3$, one-tailed Mann–Whitney (MW) test; Fig. 4C). Similar tendencies were observed in other tissue models, except for the weaker liver

and lung models (Appendix Fig. S6). A collective analysis of all tissue models based on the median ranks of each subset of features per model ("Methods"), again showed that tissue-specific features of the modeled tissues rank significantly better than variant-specific features, as well as features of other tissues (adjusted $P = 0.019$ and 0.001, respectively, paired Wilcoxon test; Fig. 4C). Thus, tissue-specificity greatly contributes to models' success.

Lastly, to substantiate that TRACEvar tissue models have common determinants, we created a multi-tissue model. To this end, we restructured our features and labeling datasets to pertain to pairs, which were composed of a variant and a tissue ("Methods"). We labeled each pair according to the pathogenicity of the given variant in the given tissue, and associated the pair with five features that were specific to the given tissue and 84 features that were specific to the given variant ("Methods"). We estimated the performance of this multi-tissue model via cross-validation, by training the model on all pairs except for pairs pertaining to a specific tissue, then testing the model on the left-out pairs ("Methods", Fig. 5A). The model achieved a median auROC of 0.935 and a median auPRC of 0.149 (expected 0.0149), and performed significantly better than CADD (Fig. 5B, $P = 9e-5$ (auROC) and $P = 0.044$ (auPRC), MW test). We then used SHAP to display the 20 top important features (Fig. 5C). Similar to the tissue models, the two topmost features belonged to the variant-specific group of pathogenic scores. These were followed by all five tissue-specific features, ranking third to seventh, with high feature values corresponding to pathogenicity. Further analysis revealed that the tissue-specific features ranked significantly better than the variant-specific features ("Methods"; $P = 2.6e-6$, one-tailed MW test, Fig. 5D). Altogether, our results show that variants share similar pathogenicity patterns across tissues, and pathogenicity highly depends on tissue contexts.

## Discussion

The etiological understanding of diseases and the genetic diagnosis of patients are highly challenging tasks, as evident from current success rates (The 100,000 Genomes Project Pilot Investigators et al, 2021). Recent approaches to address these challenges have incorporated novel types of information, such as exon expression across tissues (Cummings et al, 2020), multiplex assays of variant effects (Esposito et al, 2019), healthcare data

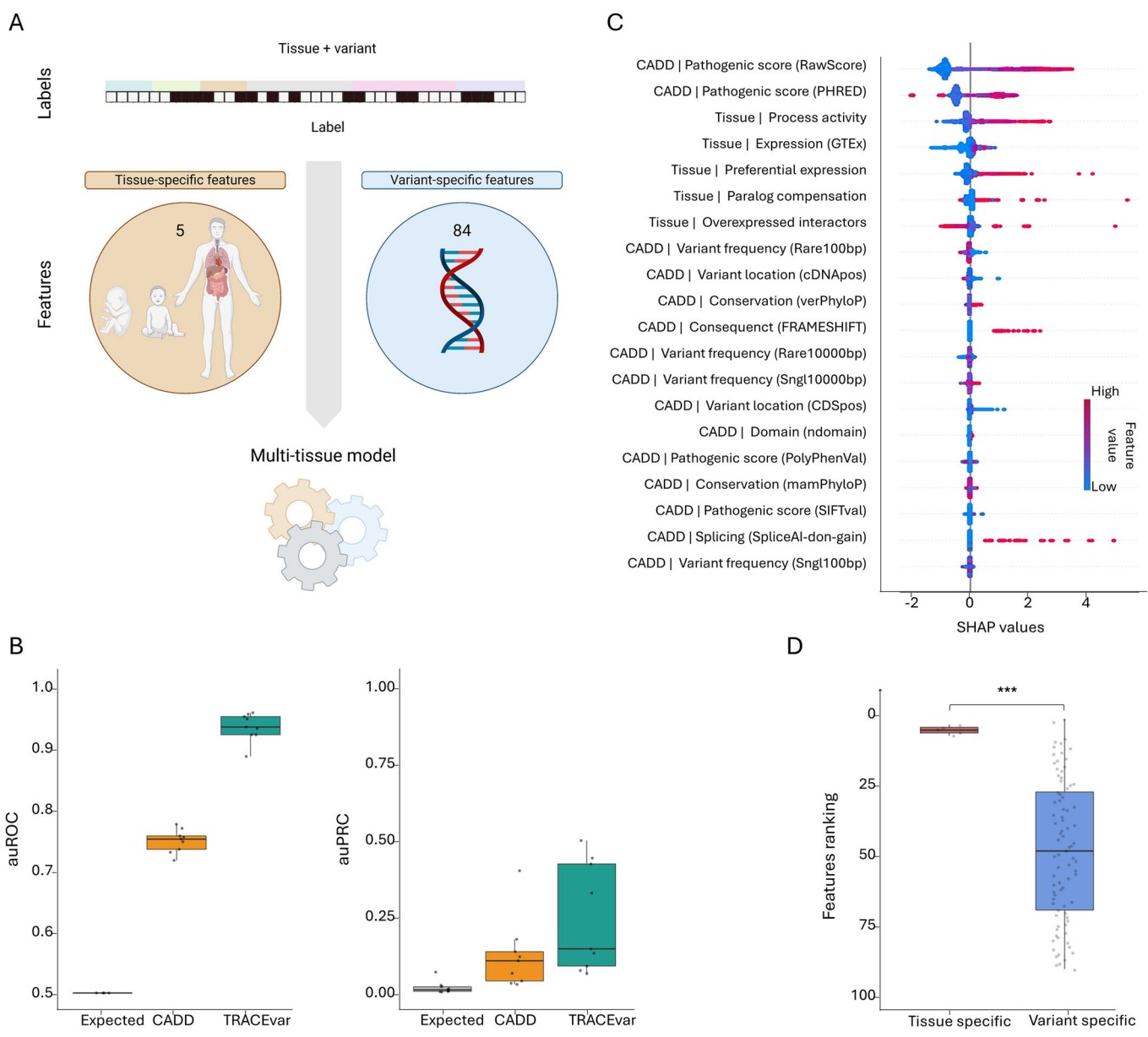

**Figure 5. A TRACEvar multi-tissue model supports the characteristics of pathogenic variants.**

(A) The multi-tissue model classified pairs comprised of a variant and a tissue. For each pair, the features dataset contained 84 variant-specific features and five tissue-specific features pertaining to the paired tissue. The model was tested by training on pairs pertaining to eight tissues, and testing on pairs pertaining to the left-out tissue. (B) The auROC (top) and auPRC (bottom) of TRACEvar multi-tissue model and CADD versus random expectation ($n = 9$). TRACEvar performed significantly better than both (one-tailed paired Wilcoxon test, $P = 0.002$). Boxplot central band indicates median; box limits indicate 25th to 75th percentiles; whiskers indicate 1.5 × interquartile range. (C) The 20 most important features of the TRACEvar multi-tissue model. Features were ordered from bottom to top by their increased absolute contribution to the model. Per feature, each dot represents the feature value of a different variant (red and blue denote high and low values of the feature, respectively). Dots are spread from left to right by their contribution to disease manifestation in the modeled tissue. (D) The ranking of variant-specific features (blue) and tissue-specific features (red) of TRACEvar multi-tissue model. Tissue-specific features ranked significantly better than variant-specific features (one-tailed paired Wilcoxon test, $P = 2.6e-6$). Boxplot representation is as described above.

(Kaplanis et al, 2020), or protein structures predicted by AlphaFold2 (Jagota et al, 2023).

Here we presented TRACEvar, a tissue-aware variant interpretation scheme that utilized the identity of disease-affected tissues. Notably, information on a patient's tissue or physiological system that is affected by a disease is routinely recorded by

clinicians and readily available for use, yet has rarely been part of the input to variant effect prediction schemes (The 100,000 Genomes Project Pilot Investigators et al, 2021; Kohler et al, 2021).

To accomplish tissue-awareness we created two unique datasets. Firstly, a tissue-aware variants dataset where 18,634 variants were

labeled by their pathogenicity in 14 tissues. Secondly, a tissue-aware features dataset consisting of 84 variant-specific features and 495 tissue-specific features, thereby dramatically shifting emphasis toward tissue contexts (Fig. 1). We tested six ML methods for TRACEvar implementation, of which GBM performed best (Appendix Fig. S2). The resulting TRACEvar models successfully distinguished pathogenic from benign variants in 14 tissue contexts (Appendix Fig. S2). Notably, when tested on recently annotated pathogenic variants, TRACEvar outperformed ten well-established variant effect prediction tools that were tissue-oblivious (Fig. 2). Hence, tissue contexts are valuable for variant effect prediction. The auPRC values, which are more relevant measures in imbalanced classification tasks as this one, correlated with the number of pathogenic variants across tissues (Appendix Fig. S2), which could imply that additional curation efforts could enhance performance.

TRACEvar implementation has limitations. Firstly, the annotation of variants pathogenicity and their association with affected tissues could be misleading or partial. To enhance the reliability of variants annotations we utilized ClinVar review status and included only variants with at least two gold stars, which led to a workable dataset size. To enhance the reliability of tissue associations, we used expert manual classification of diseases to affected tissues (Hekselman et al, 2022). Second, the performance of TRACEvar and other methods might be biased by circularity. To limit data circularity, we created disjoint training and test sets, and further restricted the test set to recently annotated variants, as these were less likely utilized for training other methods. Another form of circularity stems from the likelihood that pathogenic variants that were used for testing were themselves interpreted and curated using variant effect prediction methods. For example, CADD scores are a popular source of computational evidence for pathogenicity, and indeed pathogenic variants identified in patients were ranked highly by CADD, suggesting that CADD or a related method were used to interpret candidate variants prior to their experimental testing (Fig. EV2). This type of circularity is hard to control for, and could affect multiple methods. Yet, in support of the added value of tissue contexts, when applied to the test set TRACEvar outperformed all methods (Fig. 2C–F). We further assessed the specificity of tissue contexts by ranking pathogenic variants of each tissue using models of other tissues. These analyses showed that models of the "correct" tissues performed better than models of the "wrong" tissues (Appendix Fig. S4; Fig. EV1). Lastly, a comparison of TRACEvar to a hybrid approach suggested that simultaneous analysis of variant features and tissue contexts is better than their combination post hoc (Appendix Fig. S3). Taken together, these analyses demonstrate that tissue contexts could enhance variant effect prediction.

Whereas ML has been usefully applied for prediction tasks (Li et al, 2020; Somepalli et al, 2021), the mode of action of pathogenic variant has often remained elusive: Does the mutation affect a conserved region? Is the mutated gene expressed preferentially during embryonic development? Answers to such questions could improve disease understanding, but have not been addressed rigorously by current variant interpretation schemes. Here, we harnessed interpretable ML methods and SHAP analysis toward this goal. Given that a particularly unmet need is to illuminate the etiology of rare diseases, we tested TRACEvar's prediction and interpretation on candidate variants in 52 genetically diagnosed rare-disease patients. These cases were previously solved by well-established variant effect prediction tools, and indeed verified pathogenic variants scored high by these tools (Fig. 3). Nevertheless, TRACEvar predictions were slightly better than CADD and SIFT (Figs. 3A and EV2). Importantly, TRACEvar interpretations pointed to likely modes of action of verified pathogenic variants (Fig. 3B–D). This interpretability function is not supported by other variant effect prediction tools. By this, ML progressed from functioning as a conceptual black box toward a generator of testable hypotheses for disease etiology.

To illuminate determinants of pathogenicity at the large scale, we interpreted TRACEvar's tissue-specific models using SHAP (Fig. 4). The topmost contributing feature across models was variant pathogenicity scores of well-established tools, supporting the credibility of our interpretation scheme (Fig. 4). Other variant-specific features, such as conservation and mutation type, also ranked high across models. Features associated with regulatory factors had low contributions, potentially due to low coverage of the variants in our dataset (<20%). Fitting with the enhanced performance of TRACEvar over tissue-oblivious methods (Fig. 2), tissue-specific features played major roles in models' decisions (Fig. 4B,C). Of those, features of disease-affected tissues were more relevant than features of unaffected tissues (Fig. 4B,C). They pointed to known characteristics of disease genes in affected tissues, such as preferential expression (Lage et al, 2008), and to recently identified characteristics, such as high process activity (Sharon et al, 2022; Simonovsky et al, 2023) (Appendix Fig. S5A). These results were supported by the TRACEvar multi-tissue model (Fig. 5C), ascertaining the credibility and interpretability of the tissue-specific ML models.

TRACEvar analysis is accessible online, and includes the ranking of user-uploaded variants, and the top features contributing to model's decision per variant. The TRACEvar framework does not aim to replace common schemes for filtering variants. Instead, we recommend using it as a complementary tool for interpreting candidate variants that were not ruled out by other effect prediction tools and require in-depth analysis. While we designed TRACEvar with the notion that the affected tissue is known a priori, we showed that TRACEvar scores of variants across tissues could also be used to limit the list of likely affected tissues (Fig. EV1). TRACEvar cannot interpret non-coding genetic variants, which dominate, for example, genome-wide association studies, and is less fit for systemic diseases that involve multiple physiological systems. Future efforts could expand TRACEvar to include features of non-coding genetic variants, use multi-label learning methods, or tailor models to disease classes (Kaplanis et al, 2020).

Altogether, we present a powerful ML-based framework for prioritization and interpretation of pathogenic variants. Throughout our analyses, we found that tissue contexts greatly contribute to efficient and meaningful genetic diagnosis and call for their incorporation into variant effect prediction schemes.

## Methods

### Features dataset

We associate each variant with a set of features listed in Appendix Table S1. Tissue-specific features were extracted from TRACE (Simonovsky et al, 2023). They included gene features derived from gene expression levels in 54 adult tissues (The GTEx Consortium, 2020) and in seven embryonic tissues (Cardoso-Moreira et al, 2019), PPIs,

biological process activities across tens of tissues (Sharon et al, 2023), eQTLs (The GTEx Consortium, 2020) and paralog ratios (Jubran et al, 2020). In case the same type of feature, such as biological process activities, had distinct features in TRACE for min and max values, only the max feature was included in TRACEvar. Variants of a gene were associated with the gene's features. Next, we extracted 84 variant-specific features from CADD that were defined for v1.6 hg37 and v1.6 hg38 (Rentzsch et al, 2021) (Appendix Table S1). Non-numerical features were coded by using one-hot-vector. Missing values were replaced by CADD default values for variant-specific features, or by zero for tissue-specific features (Appendix Table S1). To simplify analyses and presentation, we further divided the 495 tissue-specific features into six main groups, such as "expression" group, and feature subgroups (e.g., "expression" and "preferential expression"; Appendix Table S1). The 84 variant-specific features were divided into 11 groups, such as the "pathogenic score" group (Appendix Table S1; Fig. 1B).

## Variant labeling

Data of annotated pathogenic and benign variants were extracted from ClinVar (Landrum et al, 2016) (Appendix Table S2). To enhance the reliability of the dataset, we included only variants with review status of two gold stars of more (i.e., criteria provided, multiple submitters, no conflicts). Of those, we included pathogenic variants that were (i) associated with OMIM disease id in the PhenotypeList column; (ii) the disease had a known molecular basis in OMIM (OMIM phenotype mapping key 3)(Amberger et al, 2009); and (iii) the disease manifested clinically in a tissue according to the ODiseA database (Hekselman et al, 2022). Specifically, tissues were considered as affected by a disease if diagnosed patients had (i) pathophysiological changes in that tissue according to clinical, laboratory, or imaging evidence; (ii) the pathophysiological change was not negligible compared to disease-related changes in other tissues, and (iii) the pathophysiological change was not documented in a small subset of the patients, as detailed in (Hekselman et al, 2022). We labeled each variant as pathogenic in the tissue affected by its respective disease and benign in other tissues. We included benign variants from ClinVar that were marked with '0' in the ClinSigSimple column and had no disease association in PhenotypeList column. These variants were labeled as benign in all tissues.

## Prediction of pathogenicity for variants in MYH6 and CLN3

We predicted variant pathogenicity in heart, whole brain, testis, lung, blood and kidney tissues by using Scikit-learn GBM model with default parameters. Per tissue, we trained a separate GBM model on all labeled variants except for variants of the predicted gene. Variants were labeled according to their pathogenicity in the modeled tissue. Next, we applied the trained model to variants of the predicted gene.

## Training and testing datasets

We divided the dataset of annotated variants (review status of two gold stars or more) into training and test sets (Appendix Table S2). The training set contained variants that were annotated in ClinVar by the year 2022. The test set contained variants that were annotated in ClinVar in the year 2023 or after. To avoid data leakage, we excluded from the test set variants in genes with other variants in the training set, if all those variants were labeled similarly (i.e., all pathogenic or all benign). We also created a limited test set that contained only variants in genes with no other variants in the training set.

## ML methods implementation and assessment

KNN, LR, RF, Adaboost, GBM, SVM, MLP Classifier were implemented by using the Scikit-learn python package (Pedregosa et al, 2011). XGB was implemented by using the xgboost python package (Chen and Guestrin, 2016). We applied each ML method to model the pathogenicity of variants in the training set. Prediction was done in the context of 14 tissues, where each tissue had pathogenic variants in at least 20 genes. The performance of each method was assessed using cross-validation. The number of folds per tissue was defined such that (i) there was no gene overlap between the training and validation folds, and (ii) the proportion of pathogenic variants was similar between folds and equal to their proportion in that tissue. Benign variants were selected randomly per fold to maintain similar proportions across folds. Consequently, the number of folds per tissue ranged from two in lung, four in liver and whole blood, to 15 in whole brain. Once determined, the same data per fold was used by all ML methods. Per method and tissue, hyper-parameters of the model were tuned to achieve higher auPRC across all variants per tissue by RandomSearch Scikit-learn function. The selected parameters per tissue model appear in GitHub (see "Data availability" section). The performance of each method per tissue was assessed by its average auROC and auPRC. We compared the performance across tissues of different methods using paired Wilcoxon test. We also measured the model training and prediction time (Appendix Fig. S2). GBM models obtained the highest auPRC and hence GBM was selected for TRACEvar implementation (Appendix Fig. S2). We applied the trained TRACEvar tissue models to the test set. The distribution of TRACEvar scores per tissue model appear in Appendix Fig. S7.

## Comparison to other variant effect prediction methods

The pathogenicity scores of variants by SIFT (Kumar et al, 2009), PolyPhen-2 HVAR (Adzhubei et al, 2013), and CADD were extracted from CADD v1.6 hg37 and included all available scores therein (Rentzsch et al, 2021). CAPICE scores were downloaded from its online webtool (https://capice.molgeniscloud.org/) (Li et al, 2020). The pathogenicity scores of other methods were downloaded from the dbNSFP 4.5 database (Liu et al, 2020). We assessed the performance of each method per tissue by scoring variants from the test set, and calculating the resulting auROC and auPRC. In case the method did not score all variants in the test set, the performance of TRACEvar was recalculated based on the subset of variants that were scored by that method. We compared the performance of TRACEvar and other methods across tissues using a paired Wilcoxon test.

## Comparison of TRACEvar to a hybrid approach

The hybrid approach ranked each variant in the test set by averaging its REVEL score and the TRACE score of its corresponding gene in the respective tissue model (TRACE scores were available for 10 tissue models). TRACEvar was then applied to these variants as described above. Analyses were done per tissue, by comparing the ranking of pathogenic

variants of that tissue between the hybrid approach and the TRACEvar tissue model using paired Wilcoxon test. We also compared ranking collectively across models, by associating each tissue with the median rank of its pathogenic variants by the hybrid approach and by TRACEvar, and then comparing these median ranks across tissues using paired Wilcoxon test.

## Additional analyses of TRACEvar models

We tested the impact of using the wrong TRACEvar tissue model. For that, per tissue  $t$ , we applied TRACEvar models of each of the other tissues to predict the labels in $t$ of variants in the test set. We then compared the auROC of the correct tissue model to the auROCs of the other tissue models. We also tested whether TRACEvar could be used to predict the tissue that is likely affected by a pathogenic variant. For that, we compared the ranks of pathogenic variants in our test set between their affected tissue and other tissues. Lastly, we tested if TRACEvar models were biased by an overlap between disease genes and biological process features derived from GO. For that, we removed those features and repeated the training and testing of TRACEvar models. We assessed the performance of tissue models with and without those features using auRPOC and auPRC, and compared their performance using paired Wilcoxon test (Appendix Fig. S8).

## Model interpretation

Feature importance values were calculated using SHAP (Lundberg et al, 2020). For each tissue-specific model, we created a SHAP explainer that calculated the impact of each feature on the prediction of a variant (denoted SHAP value). The importance of each feature was set to the median SHAP value obtained for a given tissue-specific model. To generalize feature contribution across models, we grouped features by their type (Appendix Table S1). Variant-specific features were grouped into 11 concept groups (Appendix Table S1). Tissue-specific features were divided into groups, which we defined according to ref. (Simonovsky et al, 2023) (Appendix Table S1). Each tissue-specific feature group was further divided into features of the modeled tissue and features of other tissues. For example, in the skeletal muscle tissue model, the feature "expression in skeletal muscle" was among the features of the modeled tissue, whereas the feature "expression in liver" was among the features of other tissues. To assess the contribution of each group of features to a tissue model, we normalized the contribution of each feature by the summed contribution of all features to that model. Next, we set the contribution of each group to the mean normalized contributions of its features.

## Multi-tissue model

The model was based on pairs of a variant and a tissue. Each pair $(v,t)$ was labeled according to the pathogenicity of variant $v$ in tissue $t$, and was associated with 84 variant-specific features and five tissue-specific features (Appendix Table S4). We applied this analysis to all labeled variants in nine tissue contexts. Brain regions were excluded to avoid bias due to their similarity; kidney tissue was excluded due to lack of certain tissue-specific features. A GBM model with default parameters was trained on pairs corresponding to all tissues except one and was tested on pairs corresponding to the left-out tissue. Feature importance was computed for a model that was trained on all pairs.

## Statistical analyses of the ranking of variant-specific versus tissue-specific features

Per the tissue model, we divided all features into variant-specific features, tissue-specific features of the modeled tissue (see above), and tissue-specific features of other tissues. For simplicity, differential PPIs and elevated PPIs were combined into a single group. Next, we compared the ranks of variant-specific features to the ranks of the two types of tissue-specific features using one-tailed MW test. The comparison included all features, without limiting them to the fraction of features that ranked highly. All $P$ values were adjusted for multiple hypothesis testing using the Benjamini–Hochberg procedure. We applied a similar analysis to the multi-tissue model. We also did a collective analysis across tissue models. For that, we calculated the median rank of each group of features per tissue model and then compared the median ranks of the different feature groups across all tissue models using a paired one-tailed Wilcoxon test.

## Description of clinical data

We analyzed genetic data from 52 patients with hereditary diseases with tissue-specific manifestations. The diseases of five patients manifested in two tissues, resulting in 57 cases overall (Appendix Table S3). All cases corresponded to genetic diseases with Mendelian inheritance that were investigated by the lab of one of the co-authors, Prof. Ohad Birk. All cases had (i) extensive clinical data available, (ii) underwent NGS investigations using modern techniques and subsequent analysis, as previously described (Drabkin et al, 2018; Perez et al, 2018; Wormser et al, 2019; Yogev et al, 2017), (iii) high-quality data files were available, (iv) the cases were published, (v) the disease manifested in tissues that could be analyzed by TRACEvar. Affected tissues were determined clinically, i.e., as the tissues that were most related to disease symptoms. We only included the clinical data known when genetic evaluation was performed. Data per patient was de-identified, and variants were filtered using an established filtration cascade. Briefly, we excluded variants from low-quality reads, variants within highly variable regions of the genome, variants with high prevalence in large genomic databases (allele frequency higher or equal to 0.5% in 1000 genomes project (Clarke et al, 2017), NHLBI ESP exomes (Auer et al, 2012), ExAC (Bahcall, 2016) and gnomAD (Karczewski et al, 2020)), and variants with a high prevalence in the unique studied population. We kept variants that were predicted to be possibly deleterious, such as predicted to cause loss-of-function or missense variation, alter splicing or have a CADD score >15, and are in a state of zygosity consistent with the presumed mode of inheritance.

## Analysis of clinical data by TRACEvar and other methods

Each variant was associated with its variant-specific and tissue-specific features, as described above. Per case, we constructed a TRACEvar GBM model for the patient's affected tissue. Missing values were replaced by CADD default values for variant-specific features or by zero for tissue-specific features (Appendix Table S1). The model was trained in two ways: (i) a 'full' tissue model was

trained by using all variants, and (ii) a "case-specific" tissue model was trained on all variants, except for variants in the same genes as the candidate variants of that patient, to limit data leakage. Each model was then applied to the candidate variants of the patient. The results obtained via the full tissue models were described in the main text. The results obtained via the case-specific tissue models were similar and appear in Appendix Fig. S9. We applied all tools that we previously applied to the test set to the same data per patient. Per method and case, all variants of that patient were ranked relative to each other by their pathogenicity score, and the rank of the verified pathogenic variant was recorded (Dataset EV1). In case the method did not score all variants in the test set, the ranking by TRACEvar was recalculated based on the subset of variants that were scored by that method. The ranks of pathogenic variants were compared between TRACEvar and each method across all cases using one-tailed paired Wilcoxon test.

## TRACEvar webtool

The TRACEvar webserver allows users to upload query variants and select one of 14 tissue-specific models, which were trained using the entire variants dataset. The selected model is then applied online to the query variants. The output includes variants' scores and SHAP analysis showing the model's decision plot per variant. The webserver was implemented in Python using the Flask framework. Data of tissue-specific features was stored on a MySQL database. Data of variant-specific features were extracted by running CADD V1.6 offline version. The website client was developed using the ReactJS framework and designed with Semantic-UI. The charts were displayed by the Google-Charts library. The TRACEvar webserver supports all major browsers.

## Data availability

Datasets of TRACEvar features, and TRACEvar training and testing are available at Zenodo (https://zenodo.org/records/11528443). Code for creating TRACEvar, along with the selected parameters per ML method and tissue, are available in GitHub (https://github.com/ChananArgov/TRACEvar).

The source data of this paper are collected in the following database record: biostudies:S-SCDT-10_1038-S44320-024-00061-6.

## Peer review information

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

## Acknowledgements

This study was funded by the Israel Science Foundation [401/22 to EY-L] and by a Ben-Gurion University grant [to EY-L]. Figures include illustrations adapted from BioRender.com.

## Author contributions

**Chanan M Argov**: Conceptualization; Formal analysis; Investigation; Methodology; Writing—original draft. **Ariel Shneyour**: Formal analysis; Investigation. **Juman Jubran**: Formal analysis; Investigation; Methodology; Writing—review and editing. **Eric Sabag**: Software. **Avigdor Mansbach**: Software. **Yair Sepunaru**: Software. **Emmi Filtzer**: Software. **Gil Gruber**: Software. **Miri Volozhinsky**: Software. **Yuval Yogev**: Data curation; Formal analysis. **Ohad Birk**: Data curation. **Vered Chalifa Caspi**: Methodology. **Lior Rokach**: Methodology. **Esti Yeger-Lotem**: Conceptualization; Supervision; Funding acquisition; Investigation; Methodology; Writing—original draft; Writing—review and editing.

Source data underlying figure panels in this paper may have individual authorship assigned. Where available, figure panel/source data authorship is listed in the following database record: biostudies:S-SCDT-10_1038-S44320-024-00061-6.

## Disclosure and competing interests statement

The authors declare no competing interests.

# Expanded View Figures

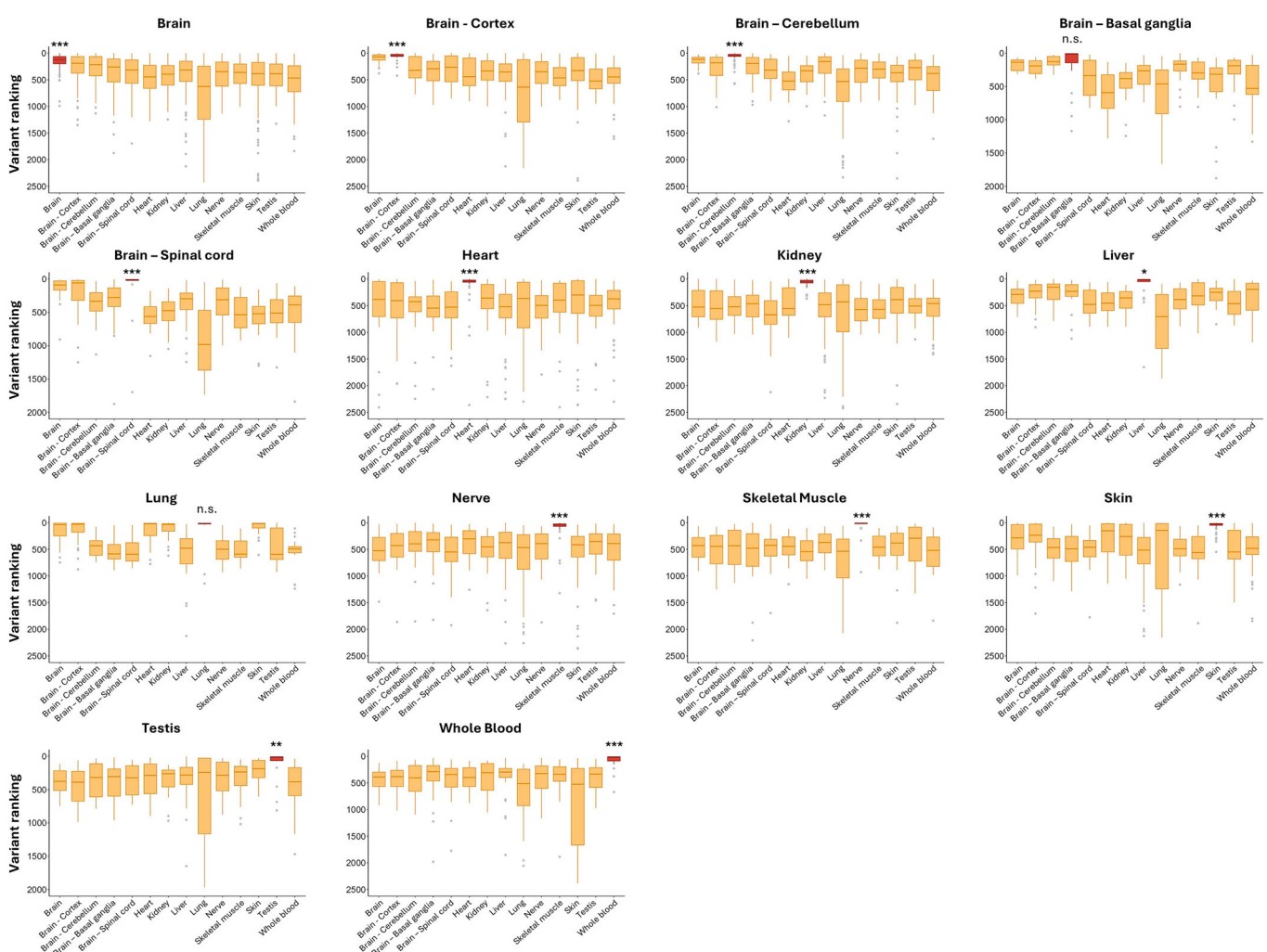

**Figure EV1.  Using TRACEvar to predict the tissue that is likely affected by a pathogenic variant.**

Pathogenic variants of an affected tissue ($19 \leq n \leq 247$) were ranked by the model of the affected tissue (red) and by each of the other tissues. In all 14 tissues, the median rank of pathogenic variants was highest when ranked by the model of their affected tissue. One-tailed paired Wilcoxon test, adjusted P values: *$P < 0.05$, **$P < 0.01$, ***$P < 0.001$, n.s. not significant. Boxplot central band indicates median; box limits indicate 25th to 75th percentiles; whiskers indicate 1.5 × interquartile range.

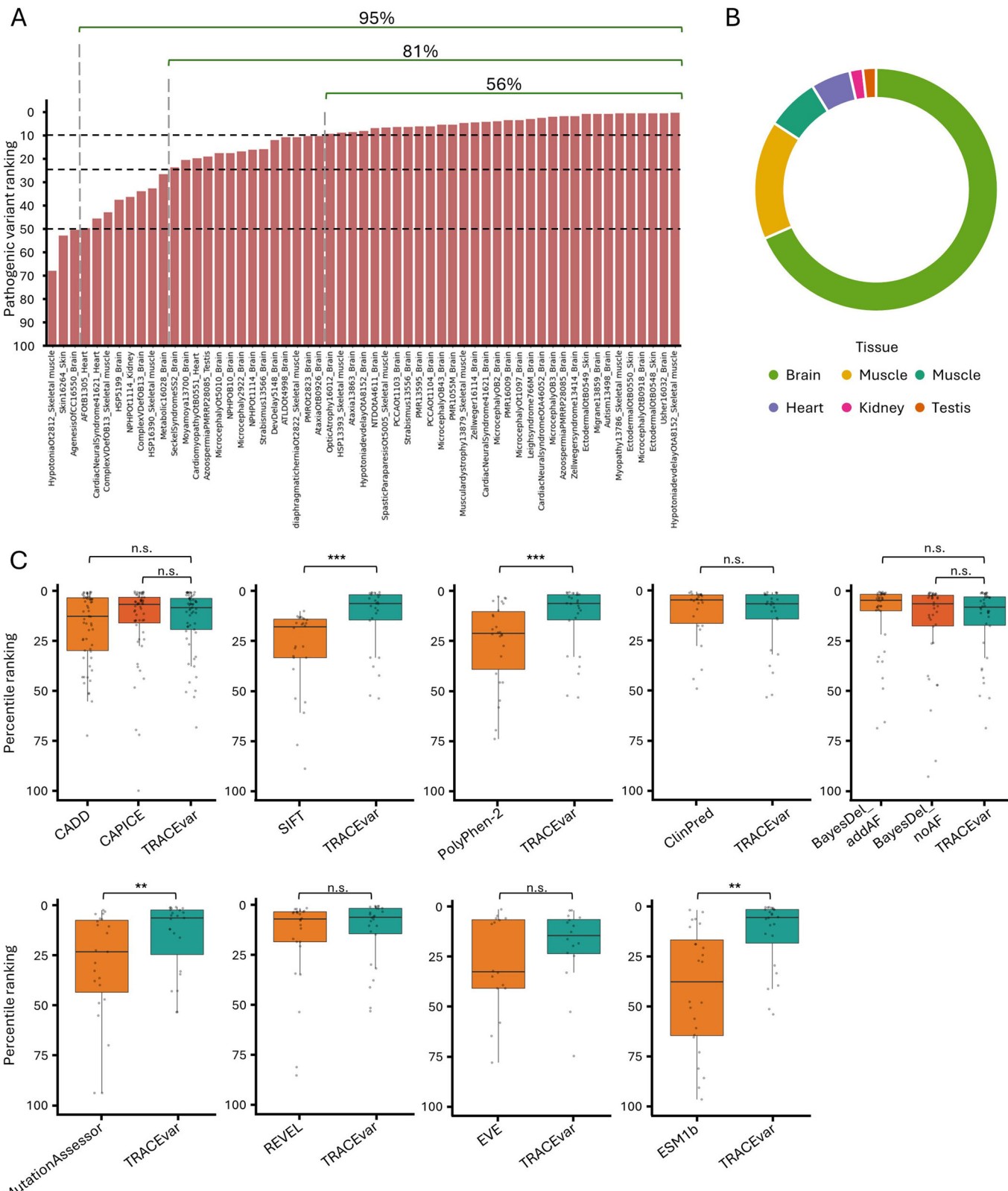

◀  **Figure EV2.   Assessment of TRACEvar performance on clinical cases.**

(**A**) The rank of the verified pathogenic variant out of the patient's candidate variants. 95% of the verified pathogenic variant ranked above the median; 81% ranked at the top quartile; 56% ranked at the top 10%, and 40% ranked at the top 5% of the variants per patient. (**B**) The distribution of cases per tissue. (**C**) Comparison between ranking of the verified pathogenic variant by TRACEvar and by other tools. TRACEvar performed significantly better than 4/10 tools, and similarly to most of the remaining tools. One-tailed paired Wilcoxon test, adjusted $P$ values: **$P < 0.01$, ***$P < 0.001$, n.s. not significant. The exact $P$ values appear in the Source Data.

    