## [Peer Review File · Molecular Systems Biology]

Tissue-aware interpretation of genetic variants advances the etiology of rare diseases

Chanan Argov, Ariel Shneyour, Juman Jubran, Eric Sabag, Avigdor Mansbach, Yair Sepunaru, Emmi Filtzer, Gil Gruber, Miri Volozhinsky, Yuval Yogev, Ohad Birk, Vered Chalifa Caspi, Lior Rokach, and Esti Yeger-Lotem

Corresponding author(s): Esti Yeger-Lotem (estiyl@bgu.ac.il)

Review Timeline:

Submission Date:	8th Jun 23
Editorial Decision:	7th Jul 23
Revision Received:	16th Jun 24
Editorial Decision:	10th Jul 24
Revision Received:	8th Aug 24
Accepted:	9th Aug 24

Editors: Maria Polychronidou and Jingyi Hou

Transaction Report:

10th Jul 2023

Manuscript Number: MSB-2023-11786

Title: Tissue-aware interpretation of genetic variants advances the etiology of rare diseases

Dear Esti,

Thank you again for submitting your work to Molecular Systems Biology. We have now heard back from the three reviewers who agreed to evaluate your study. As you will see below the reviewers raise substantial concerns, which preclude the publication of the manuscript in its current form.

The reviewers think that as it stands the main conclusions are not well supported. Nevertheless, they appreciate that the goals of the study are relevant and they provide quite constructive and detailed suggestions on how the study could be revised and extended in order to address their concerns. Taken together and considering that the reviewers acknowledge the overall relevance, we would like to offer you a chance to address the issues raised in a major revision. Without repeating all the points listed below, some of the more fundamental issues are the following:

- Further analyses and appropriate statistical tests should be included to better support the main conclusions.
- The evaluation of the models needs to be better explained and the potential issue of circularity in the approach should be addressed.
- Further comparisons to existing methods should be included.
- Reviewer #2 recommends making the method available as an 'easy to use' web tool, which would indeed enhance the impact of the study.

All issues raised by the reviewers need to be satisfactorily addressed. As you may already know, our editorial policy allows in principle a single round of major revision, so it is essential to provide responses to the reviewers' comments that are as complete as possible. I understand that the required revisions are substantive. Please feel free to contact me in case you would like to discuss in further detail any of the issues raised or if you would like to share your revision plan with me. I would be happy to schedule a call.

On a more editorial level, we would ask you to address the following points:

- Please provide a .doc version of the manuscript text (including legends for the main figures) and individual production quality figure files for the main Figures (one file per figure).
- Please include 5 keywords.
- We have replaced Supplementary Information by the Expanded View (EV format). In this case, all additional figures can be included in a PDF called Appendix. Appendix figures should be labeled and called out as: "Appendix Figure S1, Appendix Figure S2... Appendix Table S1..." etc. Each legend should be below the corresponding Figure/Table in the Appendix. Please include a Table of Contents in the beginning of the Appendix. For detailed instructions regarding expanded view please refer to our Author Guidelines: .
- Please provide a "standfirst text" summarizing the study in one or two sentences (approximately 250 characters), three to four "bullet points" highlighting the main findings and a "synopsis image" (550px width and max 400px height, jpeg format) to highlight the paper on our homepage.
- We would recommend changing the article type to "Method" given the strong methodological focus of the study.
- All Materials and Methods need to be described in the main text. We would ask you to use 'Structured Methods', our new Materials and Methods format, which is mandatory for Methods and Articles with a strong methodological focus. According to this format, the Materials and Methods section should include a Reagents and Tools Table (listing key reagents, experimental models, software and relevant equipment and including their sources and relevant identifiers) followed by a Methods and Protocols section in which we encourage the authors to describe their methods using a step-by-step protocol format with bullet points, to facilitate the adoption of the methodologies across labs. More information on how to adhere to this format as well as downloadable templates (.doc or .xls) for the Reagents and Tools Table can be found in our author guidelines: . An example of a Method paper with Structured Methods can be found here: .

- Please include a "Disclosure and Competing Interests Statement" in the main text.
- Please include a Data availability section describing how the data, code etc. have been made available. This section needs to be formatted according to the example below:
The datasets and computer code produced in this study are available in the following databases:
 - Chip-Seq data: Gene Expression Omnibus GSE46748 (<https://www.ncbi.nlm.nih.gov/geo/query/acc.cgi?acc=GSE46748>)
 - Modeling computer scripts: GitHub (<https://github.com/SysBioChalmers/GECKO/releases/tag/v1.0>)
 - [data type]: [full name of the resource] [accession number/identifier] ([doi or URL or identifiers.org/DATABASE:ACCESSION])
- The References should be formatted according to the Molecular Systems Biology reference style (i.e., ordered alphabetically and listing the first 10 authors followed by et al.)
- For data quantification: please specify the name of the statistical test used to generate error bars and P values, the number (n) of independent experiments (specify technical or biological replicates) underlying each data point and the test used to calculate p-values in each figure legend. The figure legends should contain a basic description of n, P and the test applied. Graphs must include a description of the bars and the error bars (s.d., s.e.m.).
- Molecular Systems Biology supports formal data citations in the Reference list, to cite previously published datasets. In addition to citing the original papers that reported the data, we encourage you to also cite the relevant datasets directly in the Reference list. In the text, references to datasets are included as "Data ref: Smith et al, 2001" or "Data ref: NCBI Sequence Read Archive PRJNA342805, 2017". In the Reference list, data citations are very similar to normal literature references but must be labeled with "[DATASET]" at the end of the reference. For detailed instructions please refer to our Author Guidelines .
- When you resubmit your manuscript, please download our CHECKLIST (<https://bit.ly/EMBOPressAuthorChecklist>) and include the completed form in your submission.
Please note that the Author Checklist will be published alongside the paper as part of the transparent process (<https://www.embopress.org/page/journal/17444292/authorguide#transparentprocess>).

If you feel you can satisfactorily deal with these points and those listed by the referees, you may wish to submit a revised version of your manuscript. Please attach a covering letter giving details of the way in which you have handled each of the points raised by the referees. A revised manuscript will be once again subject to review and you probably understand that we can give you no guarantee at this stage that the eventual outcome will be favorable.

Kind regards,

Maria

Maria Polychronidou, PhD
Senior Editor
Molecular Systems Biology

We realize that it is difficult to revise to a specific deadline. In the interest of protecting the conceptual advance provided by the work, we recommend a revision within 3 months (5th Oct 2023). Please discuss the revision progress ahead of this time with the editor if you require more time to complete the revisions. Use the link below to submit your revision:

IMPORTANT: When you send your revision, we will require the following items:

1. the manuscript text in LaTeX, RTF or MS Word format
2. a letter with a detailed description of the changes made in response to the referees. Please specify clearly the exact places in the text (pages and paragraphs) where each change has been made in response to each specific comment given
3. three to four 'bullet points' highlighting the main findings of your study
4. a short 'blurb' text summarizing in two sentences the study (max. 250 characters)
5. a 'thumbnail image' (550px width and max 400px height, Illustrator, PowerPoint or jpeg format), which can be used as 'visual title' for the synopsis section of your paper.
6. Please include an author contributions statement after the Acknowledgements section (see <https://www.embopress.org/page/journal/17444292/authorguide>)
7. Please complete the CHECKLIST available at (<https://bit.ly/EMBOPressAuthorChecklist>).
Please note that the Author Checklist will be published alongside the paper as part of the transparent process (<https://www.embopress.org/page/journal/17444292/authorguide#transparentprocess>).
8. When assembling figures, please refer to our figure preparation guideline in order to ensure proper formatting and readability in print as well as on screen:

See also figure legend guidelines: <https://www.embopress.org/page/journal/17444292/authorguide#figureformat>

9. Please note that corresponding authors are required to supply an ORCID ID for their name upon submission of a revised manuscript (EMBO Press signed a joint statement to encourage ORCID adoption).

(<https://www.embopress.org/page/journal/17444292/authorguide#editorialprocess>)

Currently, our records indicate that the ORCID for your account is 0000-0002-8279-7898.

Link Not Available

The system will prompt you to fill in your funding and payment information. This will allow Wiley to send you a quote for the article processing charge (APC) in case of acceptance. This quote takes into account any reduction or fee waivers that you may be eligible for. Authors do not need to pay any fees before their manuscript is accepted and transferred to the publisher.

EMBO Press participates in many Publish and Read agreements that allow authors to publish Open Access with reduced/no publication charges. Check your eligibility: <https://authorservices.wiley.com/author-resources/Journal-Authors/open-access/affiliation-policies-payments/index.html>

*** PLEASE NOTE *** As part of the EMBO Press transparent editorial process initiative (see our Editorial at <https://dx.doi.org/10.1038/msb.2010.72>), Molecular Systems Biology publishes online a Review Process File with each accepted manuscripts. This file will be published in conjunction with your paper and will include the anonymous referee reports, your point-by-point response and all pertinent correspondence relating to the manuscript. If you do NOT want this File to be published, please inform the editorial office at msb@embo.org within 14 days upon receipt of the present letter.

Reviewer #1:

The authors present TRACEvar, a new method based on their previously-published TRACE approach for identifying gene-tissue disease relationships that attempts to describe variant-tissue disease relationships. This is an interesting and potentially impactful line of research, but the study as presented has a few major flaws that hold it back.

Major comments

The study combines a large number of features and attempts to rank variants based on their likely pathogenicity in a specific tissue. Many of the measures of performance are based on a fraction of features that ranked highly (page 10, first paragraph) or the number of variants that ranked highly (page 4, second paragraph). However, all of these cutoffs seem fairly arbitrary and difficult to evaluate. Furthermore, the claim that tissue-specific features have high importance is very unsurprising when we note that 495/580 features were tissue-specific. The authors should apply appropriate statistical tests to back up these claims, and also note outliers where the model did not perform as expected.

The comparison tasks used to evaluate the models seem a bit contrived. The other models being compared do not have any awareness of tissues, so it wouldn't make sense to expect them to make tissue-specific calls. Penalizing a non-tissue-aware model for calling a variant pathogenic in a tissue context where it's not expressed seems overly harsh; this may be a pathogenic variant in other contexts. The authors should instead compare the performance of TRACEvar to a state of the art non-tissue aware model like EVE or REVEL, with an additional filtering step done using TRACE to assign tissue importance. I would imagine that, because these models use more sophisticated approaches for variant effect prediction, that this approach could outperform TRACEvar on a more level playing field.

The true pathogenic variants were taken either from ClinVar or from patient exome sequences performed by the authors. However, the authors do not acknowledge the serious problem of circularity in their approach. Their model uses, among other things, CADD scores, which are a popular source of computational evidence for pathogenicity. It is very likely that the ClinVar pathogenic variants that are used for testing were themselves interpreted using CADD or other information available to TRACEvar. The authors could have addressed this by making sure that the exome variants were not curated using data known by the model, but no aspect of this problem is mentioned in the manuscript. Additionally, the authors seem to take everything in ClinVar without regard to data quality (e.g. star rating) or other filtering. This is likely to weaken their analysis and makes it

harder to trust their results.

The model only considers a small number of variant/tissue combinations to be pathogenic, and everything else (including all variants where the gene is not an annotated disease gene!) is considered benign. It is therefore unsurprising when the model seems to predict most variants as benign. The idea that every variant we don't have an easily-mineable clinical annotation for is actually benign is simply not credible. The authors should try to add a third category of uncertain variants to their model to properly capture this, and let the true benign variants be the only benign ones. In Figure 2A and 3A we see that the maximum pathogenic score (which I assume to be the posterior probability from the random forest) is ~0.7 and most are much lower, which suggests that the model is suffering from calibration problems.

Minor comments

The last paragraph of the introduction includes some fairly detailed results that are difficult to evaluate, since we haven't yet learned what the authors actually did. This should be moved to the results section.

"Investigators et al." is caused by an incorrectly-flagged consortium author and should be corrected.

I appreciate that the authors provided the dataset and the trained model on Zenodo and GitHub.

Reviewer #2:

In this manuscript, Argov and colleagues describe a computational method and resource for the prediction of human tissue specific protein variant effects with a focus on rare disorders. The field of protein variant effect predictions is populated with many resources, some of which display impressive performance, even when compared with experimental methods to measure the effects of mutations. The key advance of this work is to combine protein variant effect predictors with information on the gene's importance for a given tissue. This gene to tissue mapping is derived from features that were previously combined in a model by the same group (Simonovsky et al. MSB 2023). At first reading, this idea itself is counter-intuitive to me. Conceptually, I tend to separate the idea of trying to predict whether a variant is likely to perturb a protein and then secondly to try to predict if the perturbation of a gene/protein or sets of genes/proteins is likely to be associated with a given organism phenotype. However, I think the approach used here does have merit. The joint prediction of a protein variant towards a specific tissue-specific phenotype seems to do better than predicting simply if a variant is deleterious.

The authors first build 17 models to predict variants with an impact on 17 different tissues. The performance in predicting left out variants is impressive, and higher than using methods based only on the variant effects. The authors then apply these models in a retrospective analysis of clinical example cases with high success. They then, explored the contributions of different features to the model, showing that several tissue-specific features are of a high importance. Finally, they build a multi-tissue-model to show that the feature importance are shared between tissues.

Overall, I think this is an important study, that moves the field of variant effect prediction towards specificity of phenotype. I may not fully agree with the conceptual idea of joining variant effect and phenotype prediction in the same prediction step but I am convinced by the results.

I have only a few minor comments/suggestions:

- One thing that was left unclear to me is whether we would need to know the affected tissue a priori before running the predictions or if the models would correctly predict the affected tissue. In other words, if I had a list of variants and wanted to predict what was going to be the affected tissue, would I get a correct tissue prediction after running all models? Could the authors test this ?
- How much worse are the predictions if we run it with the "wrong" tissue model ? What would be the average AUC of predicting variants of a tissue with the models trained for all other tissues ? I assume the predictions based on the "wrong" tissue will mislabel several pathogenic variants as benign by down-weighting high VEP scores in genes that are not relevant to that tissue.
- Depending on the answers above, there might need to be better guidance in the discussion on how best to use these models.
- One minor concern I had is that some features contain the potential for bias given that the feature and the disease list of genes could be, at least in part, overlapping. This comes to mind for GO terms and rare disorder genes. There could be a bias in that rare disorder genes are highly under-determined and it is more likely that new disease genes will be labelled if they fall within the same pathway/complex of previously known ones. I don't think this is a major concern but the authors could remove features based on GO and just show that this does not strongly affect the performance.
- The figures could use some work. Specifically, the fonts were too small to read well

- I comment the authors for sharing fully the data, code and making the method available in an easy to use web tool.

Reviewer #3:

The authors present a novel variant effect predictor, TRACEvar that is capable of making tissue-aware predictions of variant pathogenicity and also boasts greater interpretability than many other models, providing feature contributions along with the model output. The demonstrated performance of TRACEvar is very good and the tissue-specific predictions are something that would be of immense value to diagnostics and clinical research. However, we believe that there are several analyses that would improve this study, particularly during the comparison of TRACEvar to other predictors.

- The statement: "Hence, tissue-specific scoring improves the prediction of variant pathogenicity." Does not have enough evidence to back it up. SIFT and PolyPhen-2 are outdated (despite appearing in every benchmark) while CAPICE is relatively unknown. The comparison should be to the methods that TRACEvar will be competing against, modern metapredictors like ClinPred/BayesDel/MutationAssessor2 and powerful language models/unsupervised predictors like ESM-1v/-b, CPT or EVE.
- While crossvalidation is a useful tool to prevent data circularity from inflating performance estimates for TRACEvar, no mention is made of measures taken to avoid data circularity in the assessment of the other predictors (particularly relevant to PolyPhen-2 and CAPICE). While this may seem academic considering TRACEvar out-performed them anyway, ensuring that the comparison was fair is still important. Particularly if further predictors are integrated into the analysis.
- It is not mentioned which version of PolyPhen-2 is used (HumVar or HumDiv), they differ in terms of the training data.
- "Per case, we constructed a TRACEvar RF model for the patient's affected tissue." - This is not in line with how scores from the web-tool are created (I believe). The web-tool provides access to 17 pre-constructed models. The performance of models that are built per-patient, per-tissue do not tell us how useful the scores that users can get from the web tool for similar scenarios are. In other words, custom-built in-house models are being assessed here, not the same TRACEvar that I can access online. I would like to see this done with the static models available through the web tool instead.
- "PolyPhen and SIFT scores were applicable only for SNV mutations; other mutations appear as NA". Clarification on what this means is needed. PolyPhen-2 and SIFT are applicable to non-SNV single amino acid substitutions and the scores are easily accessible through the online tools or by running them locally. If this line is specifically in regard to indels then it needs to specify (although SIFT-indel also exists). This should be in the methods with full clarification rather than in a footnote in a supplementary table.
- Related to the above, if SIFT and PolyPhen-2 scores are unavailable for some of the variants used for the performance assessment, then the precision-recall curves such as those shown in Figure 2 C,D and F are not comparable as the class balance will be different between the various predictors.
- Minor: PolyPhen is used throughout to refer to PolyPhen-2. PolyPhen (decision-tree based) is a very different predictor to PolyPhen-2 (Random Forest). The paper should always be referring to it as PolyPhen-2.

Dear Dr. Maria Polychronidou and Anonymous Reviewers,

Thank you for your valuable and insightful comments on our manuscript, titled 'Tissue-aware interpretation of genetic variants advances the etiology of rare diseases'.

We revised the manuscript and methods considerably to include all the analyses and changes that were proposed by the reviewers. In particular, we did the following:

(i) Refined the variants dataset to enhance its reliability. We therefore had to retrain and retest all machine-learning models and repeat all analyses. We also conducted additional analyses and statistical tests that were suggested by the reviewers.

(ii) Explained better the evaluation of the models, and addressed the potential data circularity problem by redesigning our datasets to include a test set that was not included in the training of previously published methods, allowing for an unbiased comparison.

(iii) Extended the comparison to other tools, by comparing TRACEvar performance to six additional methods, and to a total of 10 other methods.

(iv) Revised the TRACEvar web tool, which was already part of the first submission, according to the revised datasets and models (available at <https://netbio.bgu.ac.il/tracevar/>).

All the analyses supported the value of our approach and strengthened our conclusions. The resulting manuscript is accompanied by new and revised Figures, Supplementary Figures, and Supplementary Tables.

A detailed point-by-point response to each comment is provided below. The respective changes to the manuscript were marked in blue.

We hope that you will find the revised manuscript suitable for publication in *Molecular Systems Biology*.

Thank you for your kind consideration,

Sincerely,

Professor Esti Yeger-Lotem, PhD
Head, Department of Clinical Biochemistry and Pharmacology
Faculty of Health Sciences
Ben-Gurion University of the Negev, Israel

Reviewer #1:

The authors present TRACEvar, a new method based on their previously-published TRACE approach for identifying gene-tissue disease relationships that attempts to describe variant-tissue disease relationships. This is an interesting and potentially impactful line of research, but the study as presented has a few major flaws that hold it back.

Major comments

The study combines a large number of features and attempts to rank variants based on their likely pathogenicity in a specific tissue. Many of the measures of performance are based on a fraction of features that ranked highly (page 10, first paragraph) or the number of variants that ranked highly

(page 4, second paragraph). However, all of these cutoffs seem fairly arbitrary and difficult to evaluate. Furthermore, the claim that tissue-specific features have high importance is very unsurprising when we note that 495/580 features were tissue-specific. The authors should apply appropriate statistical tests to back up these claims, and also note outliers where the model did not perform as expected.

To clarify, the performance assessments of TRACEvar and other methods relied on the standard measures of sensitivity, specificity, and precision, presented using the average area under the receiver operating characteristic curve (auROC) and the average area under the precision-recall curve (auPRC). Each measure was compared to its expected values (auROC - 0.5; auPRC - the fraction of pathogenic variants in the dataset). Comparisons between methods performance were assessed statically using a paired test. These analyses were described in Methods (page 18, first and second paragraphs), Results (page 7, subsection 'Large-scale assessment of TRACEvar in 14 tissue contexts', last paragraph), Fig. 2C-F, Fig. 5B, and Appendix Fig. S2.

Regarding the number of variants that ranked highly, we mentioned it in the Introduction with respect to the analysis of clinical data to give a sense of the capability of TRACEvar (page 4, last paragraph). We describe the statistical analysis that compares the ranking of pathogenic variants by TRACEvar and each of the 10 other methods in Methods (page 21, second paragraph), Results (page 10 second paragraph), and Fig. EV2.

Regarding the number of features that ranked highly, we added a statistical analysis that compares the rank of tissue-specific features versus variant-specific features (Methods, page 20, subsection 'Statistical analyses of the ranking of variant-specific versus tissue-specific features). We applied this analysis to all TRACEvar tissue models (page 12, last paragraph, new Fig. 4C, and new Appendix Fig. S6). We also applied the same analysis to the multi-tissue model, which contained 85 variant-specific features and only 5 tissue-specific features (page 13 second paragraph and new Fig. 5D). Across models, we observed that tissue-specific features of the modeled tissue were important. We also revised the text to describe some examples (page 12, first paragraph) and outliers (page 13, first paragraph).

The comparison tasks used to evaluate the models seem a bit contrived. The other models being compared do not have any awareness of tissues, so it wouldn't make sense to expect them to make tissue-specific calls. Penalizing a non-tissue-aware model for calling a variant pathogenic in a tissue context where it's not expressed seems overly harsh; this may be a pathogenic variant in other contexts. The authors should instead compare the performance of TRACEvar to a state of the art non-tissue aware model like EVE or REVEL, with an additional filtering step done using TRACE to assign tissue importance. I would imagine that, because these models use more sophisticated approaches for variant effect prediction, that this approach could outperform TRACEvar on a more level playing field.

To clarify, we originally compared TRACEvar to non-tissue aware models for lack of tissue-aware models. This allowed us to test whether tissue contexts can contribute to variant interpretation. Following the reviewer's advice, we compared the performance of TRACEvar to a hybrid approach based on REVEL, with an additional ranking step done using TRACE to assign tissue importance (Methods, page 18, subsection 'Comparison of TRACEvar to a hybrid approach'). We applied this analysis to model each tissue with an available TRACE model. The results, which show the value of TRACEvar, were described in Results (page 9, second paragraph), and new Appendix Fig. S3.

The true pathogenic variants were taken either from ClinVar or from patient exome sequences performed by the authors. However, the authors do not acknowledge the serious problem of

circularity in their approach. Their model uses, among other things, CADD scores, which are a popular source of computational evidence for pathogenicity. It is very likely that the ClinVar pathogenic variants that are used for testing were themselves interpreted using CADD or other information available to TRACEvar. The authors could have addressed this by making sure that the exome variants were not curated using data known by the model, but no aspect of this problem is mentioned in the manuscript. Additionally, the authors seem to take everything in ClinVar without regard to data quality (e.g. star rating) or other filtering. This is likely to weaken their analysis and makes it harder to trust their results.

We thank the reviewer for these important comments. Regarding circularity, unfortunately it was not feasible to limit the variants dataset to exome variants that were not curated using CADD or related methods. These methods have been in use for several years, and are likely to have biased even experimentally validated pathogenic variants in case those variants were themselves interpreted using CADD or other methods prior to their validation. Indeed, the analyses of patients' data suggested circularity, as pathogenic variants had very high CADD scores (Fig EV2C). Nevertheless, when applied to the test set, TRACEvar outperformed other methods (Fig. 2C-F), suggesting that tissue contexts are valuable. We revised the manuscript to acknowledge this serious problem in Discussion (page 14, bottom paragraph), and also note it in Results (page 10, second paragraph).

Regarding data quality, we followed the reviewer's advice and adopted the star rating of ClinVar. The revised variants dataset included only pathogenic and benign variants with ClinVar review status of two gold stars or more (Results, page 7, subsection 'Large-scale assessment of TRACEvar in 14 tissue contexts'; Methods, page 17, subsection 'Variant labeling'; Appendix Table S2). We applied all the analyses described in the original manuscript to this refined dataset of variants. Accordingly, we updated all the results presented in the revised manuscript, including Figs. 2-5, Appendix Figs. S1-S9, and Figs. EV1-2. The revised analyses supported and strengthened our original conclusions.

The model only considers a small number of variant/tissue combinations to be pathogenic, and everything else (including all variants where the gene is not an annotated disease gene!) is considered benign. It is therefore unsurprising when the model seems to predict most variants as benign. The idea that every variant we don't have an easily-mineable clinical annotation for is actually benign is simply not credible. The authors should try to add a third category of uncertain variants to their model to properly capture this, and let the true benign variants be the only benign ones. In Figure 2A and 3A we see that the maximum pathogenic score (which I assume to be the posterior probability from the random forest) is ~0.7 and most are much lower, which suggests that the model is suffering from calibration problems.

As described in our previous answer, we filtered the variants dataset according to the star method of Clinvar. The revised dataset included benign variants annotated with at least two gold stars. Variants that were not annotated with at least two gold stars were not included. All the results presented in the revised manuscript pertain to this refined dataset of variants.

The revised Figures 2A and 3A show that the maximum pathogenic score (which is indeed the posterior probability from the model) was close to 1 for the presented cases. We also examined the distribution of the revised pathogenic scores across tissue models. The maximum pathogenic score was close to 1 across all models (new Appendix Fig. S7), suggesting that calibration issues are minor. It should also be noted that TRACEvar was designed to rank variants in a given tissue context, rather than across tissue contexts, and hence calibration was not a prerequisite.

Minor comments

The last paragraph of the introduction includes some fairly detailed results that are difficult to evaluate, since we haven't yet learned what the authors actually did. This should be moved to the results section.

We simplified the last paragraph of the introduction by splitting it into two paragraphs that describe the results more broadly (page 4 last paragraph, page 5 first paragraph).

"Investigators et al." is caused by an incorrectly-flagged consortium author and should be corrected.

Fixed.

I appreciate that the authors provided the dataset and the trained model on Zenodo and GitHub.

Thank you.

Reviewer #2:

In this manuscript, Argov and colleagues describe a computational method and resource for the prediction of human tissue specific protein variant effects with a focus on rare disorders. The field of protein variant effect predictions is populated with many resources, some of which display impressive performance, even when compared with experimental methods to measure the effects of mutations. The key advance of this work is to combine protein variant effect predictors with information on the gene's importance for a given tissue. This gene to tissue mapping is derived from features that were previously combined in a model by the same group (Simonovsky et al. MSB 2023). At first reading, this idea itself is counter-intuitive to me. Conceptually, I tend to separate the idea of trying to predict whether a variant is likely to perturb a protein and then secondly to try to predict if the perturbation of a gene/protein or sets of genes/proteins is likely to be associated with a given organism phenotype. However, I think the approach used here does have merit. The joint prediction of a protein variant towards a specific tissue-specific phenotype seems to do better than predicting simply if a variant is deleterious.

The authors first build 17 models to predict variants with an impact on 17 different tissues. The performance in predicting left out variants is impressive, and higher than using methods based only on the variant effects. The authors then apply these models in a retrospective analysis of clinical example cases with high success. They then, explored the contributions of different features to the model, showing that several tissue-specific features are of a high importance. Finally, they build a multi-tissue-model to show that the feature importance are shared between tissues.

Overall, I think this is an important study, that moves the field of variant effect prediction towards specificity of phenotype. I may not fully agree with the conceptual idea of joining variant effect and phenotype prediction in the same prediction step but I am convinced by the results.

Thank you for this detailed and sincere feedback.

I have only a few minor comments/suggestions:

- One thing that was left unclear to me is whether we would need to know the affected tissue a priori before running the predictions or if the models would correctly predict the affected tissue. In other words, if I had a list of variants and wanted to predict what was going to be the affected tissue, would I get a correct tissue prediction after running all models? Could the authors test this ?

We thank the reviewer for this suggestion. We originally developed TRACEvar with the notion that the affected tissue is known a priori. However, we tested if we could predict what was going to be the affected tissue for a list of variants. This was done by (i) taking the pathogenic variants of an affected tissue, (ii) ranking them by each tissue model, and (iii) comparing their ranking in the model of the affected tissue to their ranking by other tissue models. We found that the ranking of pathogenic variants by the model of the affected tissue was significantly better than ranking by other tissue models (new Fig. EV1). This suggests that, although not designed for this task, TRACEvar scores might be used to at least narrow the set of likely affected tissues. We describe this analysis in Results (page 9 third full paragraph; Fig. EV1), Methods (page 19, subsection 'Additional analyses of TRACEvar models'), and Discussion (page 15 first paragraph and page 16 second paragraph).

- How much worse are the predictions if we run it with the "wrong" tissue model ? What would be the average AUC of predicting variants of a tissue with the models trained for all other tissues ? I assume the predictions based on the "wrong" tissue will mislabel several pathogenic variants as benign by down-weighting high VEP scores in genes that are not relevant to that tissue.

We computed the AUC of predicting variants of an affected tissue with the models trained for all other tissues. The different AUCs are presented in new Appendix Fig. S4. Apart from pathogenic variants affecting the lung, which had equally high AUCs in the lung and skin models, the AUCs of the affected-tissue models were better than AUCs of all other tissue models. We added this analysis to Results (page 9 third paragraph; Appendix Fig. S4), Methods (page 19, subsection 'Additional analyses of TRACEvar models'), and Discussion (page 15 first paragraph).

- Depending on the answers above, there might need to be better guidance in the discussion on how best to use these models.

We refined the guidance in the discussion on how best to use the models, mentioning the analyses and observations above (page 16 second paragraph).

- One minor concern I had is that some features contain the potential for bias given that the feature and the disease list of genes could be, at least in part, overlapping. This comes to mind for GO terms and rare disorder genes. There could be a bias in that rare disorder genes are highly under-determined and it is more likely that new disease genes will be labelled if they fall within the same pathway/complex of previously known ones. I don't think this is a major concern but the authors could remove features based on GO and just show that this does not strongly affect the performance.

The features based on GO process terms aggregated multiple processes related to the same gene, and hence did not generally overlap with a list of disease genes. To test the concern for potential bias, we removed features based on GO and repeated the training and testing of all tissue models. As shown in new Appendix Fig. S8, this did not strongly affect the performance of the models. We added this analysis to Methods (page 19, subsection 'Additional analyses of TRACEvar models').

- The figures could use some work. Specifically, the fonts were too small to read well

We revised all figures to improve readability.

- I comment the authors for sharing fully the data, code and making the method available in an easy to use web tool.

Thank you for this comment.

Reviewer #3:

*The authors present a novel variant effect predictor, TRACEvar that is capable of making tissue-aware predictions of variant pathogenicity and also boasts greater interpretability than many other models, providing feature contributions along with the model output. The demonstrated performance of TRACEvar is very good and the tissue-specific predictions are something that would be of immense value to diagnostics and clinical research. **However, we believe that there are several analyses that would improve this study, particularly during the comparison of TRACEvar to other predictors.***

- The statement: "Hence, tissue-specific scoring improves the prediction of variant pathogenicity." Does not have enough evidence to back it up. SIFT and PolyPhen-2 are outdated (despite appearing in every benchmark) while CAPICE is relatively unknown. The comparison should be to the methods that TRACEvar will be competing against, modern metapredictors like ClinPred/BayesDel/MutationAssessor2 and powerful language models/unsupervised predictors like ESM-1v/-b, CPT or EVE.

We added comparisons between TRACEvar and six other methods, including modern metapredictors and language models/unsupervised predictors (REVEL, ClinPred, BayesDel, MutationAssessor2, ESM1b, and EVE). We applied each method to the variants dataset (page 8 last paragraph, and Fig. 2), and to patients' data (page 10 second paragraph, and Fig EV2), as described in Methods (page 18, subsection 'Comparison to other variant effect prediction methods'). These comparisons generally supported the value of tissue-specific scoring.

- While crossvalidation is a useful tool to prevent data circularity from inflating performance estimates for TRACEvar, no mention is made of measures taken to avoid data circularity in the assessment of the other predictors (particularly relevant to PolyPhen-2 and CAPICE). While this may seem academic considering TRACEvar out-performed them anyway, ensuring that the comparison was fair is still important. Particularly if further predictors are integrated into the analysis.

We thank the reviewer for this important comment. To address this point, we restructured the dataset of variants and divided it into training and test sets. The training set included variants annotated up to 2022, and was used to train and validate TRACEvar models. The test set included variants annotated in 2023 and after. Since these variants were recently annotated, other methods were unlikely to use them for training, hence limiting data circularity. The performance estimates of TRACEvar and other methods were consequently made using the test set and not the training set. The results supported our original conclusions.

We revised the manuscript accordingly. The training and test set were described in Results (page 7, subsection 'Large-scale assessment of TRACEvar in 14 tissue contexts') and Methods (page 17, subsection 'Training and testing datasets'; Appendix Table S2). The performance estimates over the

test set of the different methods were described in Results (page 8, second and third paragraphs) and Fig. 2.

- It is not mentioned which version of PolyPhen-2 is used (HumVar or HumDiv), they differ in terms of the training data.

We used PolyPhen-2 HVAR, and revised the text accordingly (page 18 last paragraph).

- "Per case, we constructed a TRACEvar RF model for the patient's affected tissue." - This is not in line with how scores from the web-tool are created (I believe). The web-tool provides access to 17 pre-constructed models. The performance of models that are built per-patient, per-tissue do not tell us how useful the scores that users can get from the web tool for similar scenarios are. In other words, custom-built in-house models are being assessed here, not the same TRACEvar that I can access online. I would like to see this done with the static models available through the web tool instead.

We originally used custom-built in-house models to avoid data circularity. Following this comment, we repeated the analysis using the static models available through the web tool. The results of the revised analysis appear in the Results (page 10-11), Fig. 3, and Fig. EV2. The results of the custom-built in-house models appear in Appendix Fig. S9, and the analyses were described in Methods (page 21 first paragraph).

- "PolyPhen and SIFT scores were applicable only for SNV mutations; other mutations appear as NA". Clarification on what this means is needed. PolyPhen-2 and SIFT are applicable to non-SNV single amino acid substitutions and the scores are easily accessible through the online tools or by running them locally. If this line is specifically in regard to indels then it needs to specify (although SIFT-indel also exists). This should be in the methods with full clarification rather than in a footnote in a supplementary table.

We thank the reviewer for noting this. We extracted data for these methods from the CADD database, without further limiting them by the type of mutation. We updated the relevant table accordingly and added clarifications in Methods (page 18, subsection 'Comparison to other variant effect prediction methods').

- Related to the above, if SIFT and PolyPhen-2 scores are unavailable for some of the variants used for the performance assessment, then the precision-recall curves such as those shown in Figure 2 C,D and F are not comparable as the class balance will be different between the various predictors.

We revised Fig. 2C-F such that each comparison between TRACEvar and another method, computed based on the commonly scored variants, appears in a separate graph.

- Minor:

PolyPhen is used throughout to refer to PolyPhen-2. PolyPhen (decision-tree based) is a very different predictor to PolyPhen-2 (Random Forest). The paper should always be referring to it as PolyPhen-2.

Fixed.

10th Jul 2024

Manuscript Number: MSB-2023-11786R

Title: Tissue-aware interpretation of genetic variants advances the etiology of rare diseases

Author: Chanan Argov

Ariel Shneyour

Juman Jubran

Eric Sabag

Avigdor Mansbach

Yair Sepunaru

Emmi Filtzer

Gil Gruber

Miri Volozhinsky

Yuval Yogev

Ohad Birk

Vered Chalifa Caspi

Lior Rokach

Esti Yeger-Lotem

Dear Etsi,

Thank you for sending us your revised manuscript. We have now heard back from the three reviewers who were asked to evaluate your revised study. As you will see below, the reviewers are overall satisfied with the performed revisions and support publication. Before we can formally accept the manuscript for publication, we would ask you to address some remaining issues listed below.

1. Please address the minor issues raised by the reviewers.
2. Please remove the authors contribution section from the manuscript text.
3. There is a callout for Figure 2G, but we cannot find such a panel. Please double check.
4. Data availability: Please make sure that the datasets are publicly available upon acceptance of the manuscript. Currently, the Github link is not working.
5. Funding information: Ben-Gurion University grant is missing from the online submission system.
6. Appendix: nomenclature should be Appendix Table S1-S4 in table legends.
7. Please include the legend for Dataset EV1 as a separate tab in the Excel file.
8. I have only slightly modified the synopsis text (see attached). Please let us know in case you would like to introduce further modifications.
9. Synopsis image: the text becomes blurry when the image is adjusted to the required dimensions((550px width and 400-600 px height). Could you please provide a new image with clearer text? Additionally, in the left panel, I suggest changing the text to " Is the variant pathogenic and does it affect the selected tissue?".
10. Please address the following issues raised by our data editor:
 - Please note that the legend for figure 5d is missing in the manuscript. This needs to be rectified.
 - Please note that the exact p values are not provided in the legends of figures 2e-f; 4b; EV 2c.
 - Please indicate the statistical test used for data analysis in the legends of figures 4b; EV 2c.
 - Please note that in figures 4b; EV 2c; there is a mismatch between the annotated p values in the figure legend and the annotated p values in the figure file that should be corrected.
 - Please note that for the figure EV 1, p-values are indicated in the legends. However, comparison for the same, "" ***/**/* "" has not been represented in the figures. Please rectify this in the figures or legends as applicable.
 - Please note that the box plots need to be defined in terms of minima, maxima, centre, bounds of box and whiskers, and percentile in the legends of figures 2e-f; 3a-b; 4b-c; 5b; EV 1.
 - Please note that information related to n is missing in the legends of figures 2e-f; 3a-b; 4b-c; 5b; EV 1.

Please resubmit your revised manuscript online, with a covering letter listing amendments and responses to each point raised by the referees. Please resubmit the paper ****within one month**** and ideally as soon as possible. If we do not receive the revised manuscript within this time period, the file might be closed and any subsequent resubmission would be treated as a new manuscript. Please use the Manuscript Number (above) in all correspondence.

When you resubmit your manuscript, please download our CHECKLIST (<https://bit.ly/EMBOPressAuthorChecklist>) and include the completed form in your submission. ***Please note*** that the Author Checklist will be published alongside the paper as part of the transparent process (<https://www.embopress.org/page/journal/17444292/authorguide#transparentprocess>)

Click on the link below to submit your revised paper.

Kind regards,
Jingyi

Jingyi Hou, PhD
Scientific Editor
Molecular Systems Biology

If you do choose to resubmit, please click on the link below to submit the revision online before 9th Aug 2024.

IMPORTANT: When you send your revision, we will require the following items:

1. the manuscript text in LaTeX, RTF or MS Word format
2. a letter with a detailed description of the changes made in response to the referees. Please specify clearly the exact places in the text (pages and paragraphs) where each change has been made in response to each specific comment given
3. three to four 'bullet points' highlighting the main findings of your study
4. a short 'blurb' text summarizing in two sentences the study (max. 250 characters)
5. a 'thumbnail image' (550px width and max 400px height, Illustrator, PowerPoint or jpeg format), which can be used as 'visual title' for the synopsis section of your paper.
6. Please include an author contributions statement after the Acknowledgements section (see <https://www.embopress.org/page/journal/17444292/authorguide#manuscriptpreparation>)
7. Please complete the CHECKLIST available at (<https://bit.ly/EMBOPressAuthorChecklist>). Please note that the Author Checklist will be published alongside the paper as part of the transparent process (<https://www.embopress.org/page/journal/17444292/authorguide#transparentprocess>).
8. When assembling figures, please refer to our figure preparation guideline in order to ensure proper formatting and readability in print as well as on screen:
<https://bit.ly/EMBOPressFigurePreparationGuideline>
See also figure legend guidelines: <https://www.embopress.org/page/journal/17444292/authorguide#figureformat>
9. Please note that corresponding authors are required to supply an ORCID ID for their name upon submission of a revised manuscript (EMBO Press signed a joint statement to encourage ORCID adoption). (<https://www.embopress.org/page/journal/17444292/authorguide#editorialprocess>)
Currently, our records indicate that the ORCID for your account is 0000-0002-8279-7898.

Please click the link below to modify this ORCID:
Link Not Available

10. Include a Reagents and Tools Table as part of the Methods section, which can be downloaded from our author guidelines (<https://www.embopress.org/page/journal/17444292/authorguide#structuredmethods>)

*** PLEASE NOTE *** As part of the EMBO Press transparent editorial process initiative (see our Editorial at <https://dx.doi.org/10.1038/msb.2010.72> , Molecular Systems Biology will publish online a Review Process File to accompany accepted manuscripts. When preparing your letter of response, please be aware that in the event of acceptance, your cover letter/point-by-point document will be included as part of this File, which will be available to the scientific community. More information about this initiative is available in our Instructions to Authors. If you have any questions about this initiative, please contact the editorial office (msb@embo.org).

Reviewer #1:

The authors have done an outstanding job incorporating the reviewer feedback and I sincerely hope they agree that it has substantially improved both the work and the paper.

I have a few very minor comments, none of which should be considered to block acceptance:

I tried to submit some variants to the website, and when in manual entry mode, clicking on the + to add a row caused the tab to swap back to the VCF entry tab instantly and clear the manual entry form. This happened on both Firefox and Edge (Chromium-based). Hopefully it is an easy bug to fix.

Submitting single variants on GRCh37 seems to work fine, but using GRCh38 caused the website to throw an error.

At the bottom of page 6, the authors are missing a space between the first two OMIM numbers.

The citation for "Genomes Project Pilot Investigators" is still incorrect, now missing the leading "100,000".

Reviewer #2:

The authors have largely improved the manuscripts and responded to most of my previous concerns. In particular, I appreciate the improvements around how this tool would need to be used in reference to knowing or not the affected tissues. The authors have also expanded the tests comparing this approach with tissue-naïve methods. One minor issue to still consider is simply the figures. In particular figure 3 has text that can't be read even when printing the figure in a full A4, much less if the figure is formatted to fit a journal article figure. Several other figures will have similar issues.

I think overall I think this is an interesting and important advance in our capacity to predict tissue specific effects of protein variants.

Reviewer #3:

The authors have appear to have done a good job in responding the reviewers' comments, and have made some substantial revisions to their manuscript in terms of text and analyses. In particular, they have addressed my concerns regarding comparisons to other predictors. I think that, fundamentally, making fair comparisons between methods is very challenging, but this approach provides a unique perspective that could potentially have considerable utility, and so I am supportive of its publication.

Dear Dr. Jingyi Hou,

I wish to thank you and the anonymous reviewers again for investing effort in our revised manuscript. This has been an excellent review process that allowed us to clarify, strengthen, and improve our work.

We have fixed all the remaining issues, as detailed below. Changes to the manuscript appear in blue font.

With best wishes,

Esti

Editor comments

Thank you for sending us your revised manuscript. Maria has taken on a new position at EMBO and is no longer working at Molecular Systems Biology. Therefore, I have taken over handling your manuscript. We have now heard back from the three reviewers who were asked to evaluate your revised study. As you will see below, the reviewers are overall satisfied with the performed revisions and support publication. Before we can formally accept the manuscript for publication, we would ask you to address some remaining issues listed below.

1. Please address the minor issues raised by the reviewers.

See detailed responses below.

2. Please remove the authors contribution section from the manuscript text.

Done.

3. There is a callout for Figure 2G, but we cannot find such a panel. Please double check.

Fixed, callout removed.

4. Data availability: Please make sure that the datasets are publicly available upon acceptance of the manuscript. Currently, the Github link is not working.

Checked, it is working from our end.

5. Funding information: Ben-Gurion University grant is missing from the online submission system.

Done.

6. Appendix: nomenclature should be Appendix Table S1-S4 in table legends.

Done.

7. Please include the legend for Dataset EV1 as a separate tab in the Excel file.

Done.

8. I have only slightly modified the synopsis text (see attached). Please let us know in case you would like to introduce further modifications.

Thank you, I accepted and slightly revised, see the uploaded file.

9. Synopsis image: the text becomes blurry when the image is adjusted to the required dimensions(550px width and 400-600 px height). Could you please provide a new image with clearer text? Additionally, in the left panel, I suggest changing the text to " Is the variant pathogenic and does it affect the selected tissue?".

We fixed the text and provide the image in multiple formats.

10. Please address the following issues raised by our data editor:

- Please note that the legend for figure 5d is missing in the manuscript. This needs to be rectified.

Legend added, thank you.

- Please note that the exact p values are not provided in the legends of figures 2e-f; 4b; EV 2c.

Since there are many p values, we used the stars notation. We also revised the legend by adding that "The exact p values appear in the Source Data".

- Please indicate the statistical test used for data analysis in the legends of figures 4b; EV 2c.

Done.

- Please note that in figures 4b; EV 2c; there is a mismatch between the annotated p values in the figure legend and the annotated p values in the figure file that should be corrected.

Fixed.

- Please note that for the figure EV 1, p-values are indicated in the legends. However, comparison for the same, "" ***/**/* "" has not been represented in the figures. Please rectify this in the figures or legends as applicable.

Fig. EV1 was fixed.

- Please note that the box plots need to be defined in terms of minima, maxima, centre, bounds of box and whiskers, and percentile in the legends of figures 2e-f; 3a-b; 4b-c; 5b; EV 1.

Fixed.

- Please note that information related to *n* is missing in the legends of figures 2e-f; 3a-b; 4b-c; 5b; EV 1.

Fixed. In Fig. 3A the number of patients is explicit in the text.

Reviewer #1:

The authors have done an outstanding job incorporating the reviewer feedback and I sincerely hope they agree that it has substantially improved both the work and the paper.

We thank the reviewer again, the comments indeed helped us to substantially improve both the work and the paper.

I have a few very minor comments, none of which should be considered to block acceptance:

I tried to submit some variants to the website, and when in manual entry mode, clicking on the + to add a row caused the tab to swap back to the VCF entry tab instantly and clear the manual entry form. This happened on both Firefox and Edge (Chromium-based). Hopefully it is an easy bug to fix.

Fixed.

Submitting single variants on GRCh37 seems to work fine, but using GRCh38 caused the website to throw an error.

Fixed.

At the bottom of page 6, the authors are missing a space between the first two OMIM numbers.

Fixed.

The citation for "Genomes Project Pilot Investigators" is still incorrect, now missing the leading "100,000".

Fixed.

Reviewer #2:

The authors have largely improved the manuscripts and responded to most of my previous concerns. In particular, I appreciate the improvements around how this tool would need to be used in reference to knowing or not the affected tissues. The authors have also expanded the tests comparing this approach with tissue-naive methods. One minor issue to still consider is simply the figures. In particular figure 3 has text that can't be read even when printing the figure in a full A4, much less if the figure is formatted to fit a journal article figure. Several other figures will have similar issues.

Fixed, and relevant features appear in bold. We also added the SHAP importance values as a new Dataset EV3, which is mentioned in Fig. 3 legend.

I think overall I think this is an interesting and important advance in our capacity to predict tissue specific effects of protein variants.

Thank you for the insightful and valuable comments and feedback.

Reviewer #3:

The authors have appear to have done a good job in responding the reviewers' comments, and have made some substantial revisions to their manuscript in terms of text and analyses. In particular, they have addressed my concerns regarding comparisons to other predictors. I think that, fundamentally, making fair comparisons between methods is very challenging, but this approach provides a unique perspective that could potentially have considerable utility, and so I am supportive of its publication.

Thank you for the insightful and valuable comments and feedback.

9th Aug 2024

Manuscript number: MSB-2023-11786RR

Title: Tissue-aware interpretation of genetic variants advances the etiology of rare diseases

Dear Etsi,

Thank you again for sending us your revised manuscript. We are now satisfied with the modifications made and I am pleased to inform you that your paper has been accepted for publication.

Kind regards,
Jingyi

Jingyi Hou, PhD
Scientific Editor
Molecular Systems Biology
